# Distilling Robust and Non-Robust Features in Adversarial Examples by Information Bottleneck

Junho Kim[*], Byung-Kwan Lee[*], Yong Man Ro[†]

School of Electrical Engineering
Korea Advanced Institute of Science and Technology (KAIST)
{arkimjh, leebk, ymro}@kaist.ac.kr

## Abstract

Adversarial examples, generated by carefully crafted perturbation, have attracted considerable attention in research fields. Recent works have argued that the existence of the robust and non-robust features is a primary cause of the adversarial examples, and investigated their internal interactions in the feature space. In this paper, we propose a way of explicitly distilling feature representation into the robust and non-robust features, using Information Bottleneck. Specifically, we inject noise variation to each feature unit and evaluate the information flow in the feature representation to dichotomize feature units either robust or non-robust, based on the noise variation magnitude. Through comprehensive experiments, we demonstrate that the distilled features are highly correlated with adversarial prediction, and they have human-perceptible semantic information by themselves. Furthermore, we present an attack mechanism intensifying the gradient of non-robust features that is directly related to the model prediction, and validate its effectiveness of breaking model robustness.

## 1 Introduction

Deep neural networks (DNNs) have achieved remarkable performances in a wide variety of machine learning tasks. Despite the breakthrough outcomes, DNNs are easily fooled from adversarial attacks, with crafted perturbations [1, 2, 3, 4, 5, 6, 7]. These perturbations are imperceptible to human eyes, but simply adding them to clean images (*i.e.,* adversarial examples) can effectively deceive classifiers. Such a vulnerability affects security problems [8, 9, 10, 11], bringing in the weak reliability of DNNs.

Previous works have broadly investigated the reason for the widespread of such adversarial examples. Goodfellow *et al.* [2] have argued that adversarial vulnerability is induced from the excessive linearity nature of DNNs in high-dimensional spaces. Several works [1, 12] have regarded the primary cause of the adversarial examples as statistical variation with aberrations in data manifold. Schmidt *et al.* [13] have suggested that the pervasiveness of the examples should not be considered as a drawback of training methods for DNNs, since the available dataset may not be large enough to train them robustly.

In recent years, Tsipras *et al.* [14] have suggested an intriguing analysis that the disagreement between standard and adversarial accuracy stems from differently trained feature representation. In this literature, Ilyas *et al.* [15] further have demonstrated the adversarial examples are inevitable results of standard supervised training and arisen from well-generalized features in the dataset. They have analyzed the adversarial examples are originated from brittle and unintelligible features (*i.e.,* non-robust features) that are arbitrarily manipulated with the imperceptible noise, and shown that the

---

[*]Equal contribution. [†] Corresponding author.

35th Conference on Neural Information Processing Systems (NeurIPS 2021).

robust features still can provide precise accuracy even in the existence of adversarial perturbation. They have argued that the non-robust features cannot show reliable accuracy in the adversarial setting and could provoke incomprehensible properties.

Nonetheless, the underlying reason for the existence and pervasiveness of adversarial examples cannot derive common consensus in the research field and still remains unclear [16]. To clarify where the adversarial brittleness truly comes from, we need to figure out how the robust and non-robust features in data manifold subtly manipulate feature representation and fool model prediction, by directly handling them in the feature space. To address it, we propose a way to precisely distill intermediate features into robust and non-robust features by employing Information Bottleneck (IB) [17, 18, 19]. In the sense that semantic information is included in the units of intermediate feature representation [20, 21, 22, 23], we utilize the bottleneck to regulate the information flow in the feature space by explicitly adding noise to the feature units. Then, we estimate how each feature unit contaminated with the noise affects model prediction with assigned information. Based on the prediction sensitivity of the noise intervention, we assort the feature units either robust or brittle, and disentangle the feature representation into robust or non-robust features, respectively.

Through extensive analysis of the distilled features, we corroborate that the pervasiveness of the adversarial brittleness is derived from the non-robust features, and they have a high correlation with the adversarial prediction. In addition, in order to understand the semantic information of distilled features, we directly visualize them in the feature space and provide their visual interpretation. Consequently, we reveal that both of the robust and non-robust features indeed have semantic information in terms of human-perception by themselves. Based on our observation, we theoretically describe the negative impact of the non-robust features for the model prediction and introduce an approach of amplifying the gradients of non-robust features to break the model prediction.

In this paper, our contributions can be summarized into three-fold as follows:

- We propose a novel way to explicitly distill intermediate features into the robust and non-robust features using Information Bottleneck, and interpret the disentangled features in terms of human-perception by directly visualizing them in the feature space.

- By analyzing how the distilled features affect the intermediate feature representation under adversarial perturbation, we demonstrate that the non-robust features are highly correlated with the adversarial prediction.

- We present an attack mechanism manipulating the non-robust features by strengthening their gradients, and validate its effectiveness of breaking model prediction.

## 2 Distilling Robust and Non-robust Features in Intermediate Feature Space

**Problem Setup and Notations**. Let $X$ denote clean images and $Y$ denote (one-hot encoded) target labels corresponding to the clean images. Then, adversarial examples $X_{adv}$ can be created by the following equations: $\max_{\delta} E_{(X,Y)}[\mathcal{L}(f(X + \delta), Y)]$, where $\delta$ denotes an adversarial perturbation, and $\mathcal{L}$ denotes a pre-defined loss for machine learning tasks. The adversarial examples can be made by $X_{adv} = X + \delta$. When a given model $f$ is adversarially trained against PGD attack [7], it can be written as follows:

$$\min_{w} \max_{\|\delta\|_{\infty} \leq \gamma} E_{(X,Y)} \left[ \mathcal{L}\left(f(X + \delta), Y\right) \right], \tag{1}$$

where $w$ represents the parameters of $f$, which are learned to be robust against adversarial attacks. Here, $\|\cdot\|_{\infty} \leq \gamma$ describes $L_{\infty}$ norm, and $\gamma$-ball means the perturbation magnitude. In this paper, we adversarially train the model $f$ on $\gamma = 0.03$ for the standard adversarial attack. Note that once adversarially trained, the parameters of the model $f$ are no longer covered.[1]

As notation of variables that we will use in this paper, $Z$ and $\bar{Z}$ indicate the intermediate features of the model $f$ such that $Z = f_l(X)$ and $\bar{Z} = f_l(X_{adv})$, where $f_l(\cdot)$ describes $l$-th layer outputs of the given model. Similarly, $f_{l+}(\cdot)$ represents subsequent network after the $l$-th layer, thus intermediate features can be propagated to the last output layer, such that $\hat{Y} = f_{l+}(Z)$ and $\hat{Y}_{adv} = f_{l+}(\bar{Z})$. The

---

[1]Previous works [14, 15] have demonstrated that the distinction between the robust and non-robust features arises in adversarial settings. In the sense that adversarially trained networks learn robust representation [24], we set the robust classifier as default. Please see the analysis of the standard training in Appendix F.

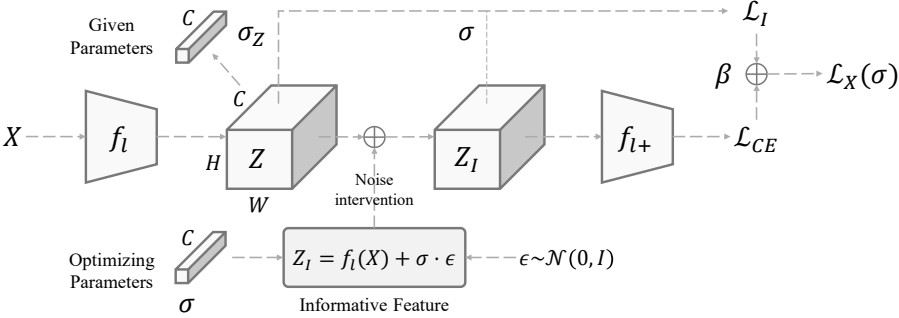

Figure 1: Diagrams of our Information Bottleneck (IB), minimizing $\mathcal{L}_{CE} + \beta\mathcal{L}_I$ to find noise variation $\sigma$ that can estimate information flow in the intermediate features $Z$. Here, $\sigma_z$ represents nature feature variation of intermediate features $Z$ for each unit, and $\epsilon$ indicates Gaussian noise sampled from $\mathcal{N}(0, I)$. More implementation details are described in Appendix A.

model $f$ can be expressed as $f = f_{l+} \circ f_l$, satisfying $\hat{Y} = f(X)$ for the given clean images $X$. Also, $\hat{Y}_{adv} = f(X_{adv})$ is denoted by model propagation of the adversarial examples $X_{adv}$. Note that we designate $l$-th layer as the last convolutional layer, and regard $l+$ as the rest of the layers in the model.

## 2.1 Information Bottleneck for Distilling Informative Features

In adversarial settings, we focus on separating robust and non-robust features in the intermediate layer. Recall that robust features are literally robust on the noise (variation) and invariant to the existence of the adversarial perturbation, but non-robust features are not. Our approach aims to distill feature units that affect target prediction under the noise perturbation in the intermediate feature space.

We follow that the semantic information is inherently included in the feature units of DNNs [20, 22, 25]. From this perspective, we utilize Information Bottleneck (IB) to distill the robust and non-robust features on the given intermediate features $Z$. Information Bottleneck [17, 18] proposed to encode maximally informative representation for target labels, restraining input information, concurrently. Using the bottleneck, we suggest a way to assess feature importance and quantify information flow for the target prediction. The objective function of IB can be written as follows:

$$\max_Z I(Z, Y) - \beta I(Z, X), \tag{2}$$

where $I$ denotes mutual information, and $\beta$ represents the degree of restraining input information. The first term $I(Z, Y)$ allows the intermediate features $Z$ to be predictive on the target label $Y$, and the second term $I(Z, X)$ encourages $Z$ to compress the information of the given images $X$ in the bottleneck. Here, the second term requires a true feature probability $p(Z) = \int_X p(Z \mid X)p(X)dX$ to expand it, but it is computationally intractable due to a high dimensional dependency of the dataset probability $p(X)$. Thus, several works [18, 19] modified the IB's objective function to make it possible to learn DNNs without the true feature probability as follows: $\min \mathcal{L}_{CE} + \beta\mathcal{L}_I$ (see Appendix B). In this formulation, $\mathcal{L}_{CE}$ indicates cross-entropy loss, and $\mathcal{L}_I$ represents information loss computed by KL divergence [26] between a feature likelihood $p(Z \mid X)$ and an approximate feature probability $q(Z)$. It is radically a closed-form approximation for the true feature probability $p(Z)$.

Firstly, we deliberately inject noise variation into $Z$ to estimate the prediction sensitivity of each feature unit along the noise intervention. To do so, we newly design an approximate feature probability using noise variation $\sigma$, such that $q_\sigma(Z) = \mathcal{N}(f_l(X), \sigma^2)$. Then, we sample random variables from $q_\sigma(Z)$ and define informative features $Z_I$ as follows:

$$Z_I = f_l(X) + \sigma \cdot \epsilon, \tag{3}$$

where $Z_I \sim q_\sigma(Z)$. Note that the operator $\cdot$ denotes Hadamard product, and $\epsilon$ stands for Gaussian noise sampled from $\mathcal{N}(0, I)$. Here, the noise variation measures a correlation between intermediate features and model prediction based on the fact that robustness means high correlation on model prediction and non-robustness are opposite [14, 15] in adversarial settings. Since the correlation $E_{(X,Y)}[Y \cdot f(X)]$ in output layer can be expressed as a variance measure $\text{Cov}(Y, f(X))$, we consider the correlation in intermediate layer as the noise variation. From this perspective, if a feature unit is

highly predictive despite large noise variation (high correlation), the unit can robustly predict target labels, while a non-robust unit cannot.

Once the informative features $Z_I$ are acquired from the noise variation $\sigma$, we propagate $Z_I$ to the last output layer and estimate the feature importance of each unit for model prediction. Then, we deal with the information loss $\mathcal{L}_I$ in order to alleviate feature heterogeneity between $Z$ and $Z_I$. Through aforementioned modification [18, 19], our objective function can be written as follows:

$$\min_{\sigma} \mathcal{L}_X(\sigma) = \underbrace{-Y \log f_{l+}(f_l(X) + \sigma \cdot \epsilon)}_{\mathcal{L}_{CE}} + \beta \underbrace{D_{KL}[p(Z \mid X) \mid\mid q_\sigma(Z)]}_{\mathcal{L}_I}, \qquad (4)$$

where the feature likelihood $p(Z \mid X)$ is set to $\mathcal{N}(f_l(X), \sigma_z^2)$. Here, $\sigma_z$ indicates inherent feature variation of the intermediate features $Z$ for each unit. The second term $\mathcal{L}_I$ makes the informative features $Z_I$ resemble the intermediate features $Z$, while minimizing the cross-entropy $\mathcal{L}_{CE}$. This second term can be written as $\mathcal{L}_I = \frac{1}{2} \sum_{k=1}^{C} [\frac{\sigma_{z_k}^2}{\sigma_k^2} + \log \frac{\sigma_k^2}{\sigma_{z_k}^2} - 1]$, where $k$ denotes an index of the noise variation $\sigma = [\sigma_1, \sigma_2, \cdots, \sigma_C]$ (the optimizing parameters) and the feature variation $\sigma_z = [\sigma_{z_1}, \sigma_{z_2}, \cdots, \sigma_{z_C}]$ (the given parameters). Here, only of the variation $\sigma$ is updated such that $\sigma \leftarrow \sigma - \frac{\partial}{\partial \sigma} \mathcal{L}_X(\sigma)$. In brief, we summarize overall procedure of our bottleneck concept in Fig. 1.

Moreover, we mention that $\beta$ in Eq. (4) controls the amount of information that flows into the feature representation. Specifically, when $\beta$ is set to zero, IB loss is equivalent to cross-entropy loss, which means that $Z_I$ can accommodate even unimportant features to predict target labels. In contrast, excessively large $\beta$ only focuses on compressing input information, thus IB may cannot filter out important features to predict target labels. Accordingly, we empirically control $\beta$ to distill informative features $Z_I$ based on the noise variation $\sigma$ (Please see section 3.4 for analysis of information flow).

## 2.2 Separating Informative Features by Tolerance of Feature Variation

After optimizing the informative features $Z_I$, we compare the noise variation $\sigma$ for the informative feature units and dichotomize each unit either robust or non-robust based on their prediction sensitivity. We set the criterion for comparison as $T = \max(\sigma_z^2)$. Here, $T$ represents the maximum tolerance of the noise variation. It is a reasonable choice to set $T$ as a criterion, because it indicates the maximal variation with respect to the changes of the given image $X$, in the feature space. In the following procedure, we explicitly disentangle intermediate features $Z$ into the robust $Z_r$ and non-robust features $Z_{nr}$.

Firstly, once the noise variation $\sigma$ is larger than the maximum tolerance $T$ in specific units, it indicates that their corresponding features are highly predictive on the model prediction, despite the noise intervention. Thus, we define their conjunction as robust features. On the other hand, if the variation of a specific unit is smaller than $T$, their corresponding features can be represented as non-robust features. This is because the small variation behaves as a strict restriction to retain model prediction of target labels. We assume that once a strong adversarial perturbation comes in, the feature variation of non-robust features becomes to be larger than acceptable tolerance, thereby easily breaking the model classifier and leading to misclassified prediction. In this respect, we sort the noise variation according to their magnitude, and cluster them by assigning robust or non-robust channel indexes to each feature unit. The robust channel index, $i_r = [i_{r_1}, i_{r_2}, \cdots, i_{r_C}]$ can be computed as follows:

$$i_{r_k} = \mathbb{1}(\sigma_k^2 > T) = \begin{cases} 1 & \sigma^2 > T \\ 0 & \sigma^2 \leq T \end{cases}, \qquad (5)$$

where $\mathbb{1}(\cdot)$ represents the indicator function. The non-robust channel index $i_{nr}$ is simply reversed from the robust channel index such that $i_{nr_k} = 1 - i_{r_k}$. Then, we estimate robust features $Z_r$ by multiplying the robust channel index to the intermediate features element-wisely such that $Z_r = i_r \cdot Z$. Similarly, non-robust features $Z_{nr}$ are presented as $Z_{nr} = i_{nr} \cdot Z$. In this way, the intermediate features $Z$ are fully disentangled into the two types of feature representation satisfying $Z = Z_r + Z_{nr}$.

To sum it up, we regard the robust features $Z_r$ that have the larger noise variation as invariant features from the adversarial perturbation. On the other hand, non-robust features $Z_{nr}$ are considered as easily manipulated features, which harmonize the smaller noise variation. Now, we analyze their impacts to the robustness by expanding the model prediction of $Z_I$ to Taylor approximation (see Appendix C.)

Table 1: Classification accuracy of model performance attacked by FGSM [2], PGD [7], and CW [4] on VGG-16 [29] and WRN-28-10 [30], adversarially trained with $\gamma = 0.03$ for CIFAR-10, SVHN, and Tiny-ImageNet. We selectively propagate each feature (*i.e.,* intermediate features (Int.), robust (R.), and non-robust features (NR.)) to measure classification accuracy.

| Model | Example | CIFAR-10 | | | SVHN | | | Tiny-ImageNet | | |
|---|---|---|---|---|---|---|---|---|---|---|
| | | Int. Acc | R. Acc | NR. Acc | Int. Acc | R. Acc | NR. Acc | Int. Acc | R. Acc | NR. Acc |
| VGG | Clean | 79.73 | 99.87 | 34.82 | 90.35 | 99.76 | 57.09 | 33.98 | 83.11 | 7.50 |
| | FGSM [2] | 51.28 | 99.58 | 22.82 | 63.71 | 98.72 | 40.12 | 17.45 | 77.74 | 4.94 |
| | PGD [7] | 44.71 | 99.38 | 20.64 | 48.92 | 97.91 | 32.18 | 16.13 | 77.59 | 4.69 |
| | CW [4] | 40.32 | 99.85 | 13.66 | 33.26 | 99.60 | 16.75 | 12.00 | 75.69 | 4.03 |
| WRN | Clean | 82.56 | 98.66 | 44.67 | 93.53 | 99.44 | 70.43 | 43.13 | 96.35 | 6.07 |
| | FGSM [2] | 56.43 | 96.80 | 31.65 | 73.93 | 97.90 | 53.96 | 20.38 | 91.58 | 3.01 |
| | PGD [7] | 51.63 | 96.61 | 29.44 | 61.09 | 96.33 | 45.79 | 18.84 | 90.37 | 2.83 |
| | CW [4] | 45.47 | 97.74 | 17.28 | 40.61 | 97.58 | 22.78 | 13.51 | 95.78 | 2.06 |

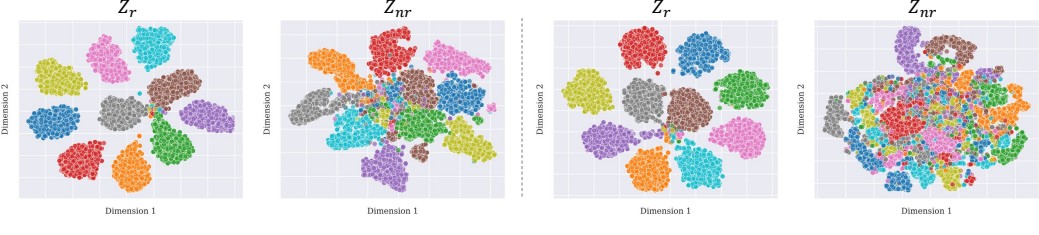

$Z_r$  $Z_{nr}$  $\bar{Z}_r$  $\bar{Z}_{nr}$

(a) Clean Examples         (b) PGD Examples

Figure 2: The result of t-SNE plot [31] in CIFAR-10 dataset for VGG-16 network. Each cluster indicates high-dimensional distributions of feature representation for 10 object labels in CIFAR-10 dataset. Additional t-SNE results for other adversarial attacks are illustrated in Appendix D.

with its convergence of local minima [27, 28] as follows:

$$f_{l+}(f_l(X) + \sigma \cdot \epsilon) = f_{l+}(f_l(X) + \sigma_r \cdot \epsilon) + \underbrace{\left[ \frac{\partial}{\partial \sigma_r} f_{l+}(f_l(X) + \sigma_r \cdot \epsilon) \right]^T \sigma_{nr}}_{\Delta}, \qquad (6)$$

where robust noise variation $\sigma_r = i_r \cdot \sigma$ and non-robust noise variation $\sigma_{nr} = i_{nr} \cdot \sigma$. Since the variation of the robust features does not degrade the model prediction, when $\sigma_{nr}$ is small enough, the erroneous term $\Delta$ in Eq. (6) closes to zero. That is, the model retains having significant robustness against the adversarial perturbation interrupting robust channel index in $Z$. Conversely, once we force $\sigma_{nr}$ to increase, its output becomes inaccurate for the robust prediction (*i.e.,* first term in Eq. (6)) due to high $\Delta$. Here, we theoretically demonstrate how the brittleness of non-robust features affects the model robustness. We will thoroughly analyze the properties of the two distilled features by empirically showing the robustness of $\bar{Z}_r$ and brittleness of $\bar{Z}_{nr}$ in the following sections.

## 3 Analysis of Distilled Features and Visual Interpretation

### 3.1 Property of Distilled Feature Units under Adversarial Perturbation

After we distill the robust and non-robust features using the bottleneck concept, our next question is *How will the target prediction change under the adversarial attacks?* In our posit, if the bottleneck successfully disentangles the robust and non-robust channel index (*i.e.,* $i_r$ and $i_{nr}$) from the given examples, we should identify the consequential classification accuracy changes under the adversarial perturbation. That is, after applying the robust index to the attacked feature representation, the selected adversarial features with $i_r$ denoted by $\bar{Z}_r$ (*i.e.,* $\bar{Z}_r = i_r \cdot \bar{Z}$) should have invariant accuracy changes for the target labels. Here, $i_r$ is robust channel index obtained from the given clean examples $X$, and $\bar{Z}$ represents the intermediate features of the adversarial examples, which means $\bar{Z} = f_l(X_{adv})$. Contrarily, the selected features satisfying $\bar{Z}_{nr} = i_{nr} \cdot \bar{Z}$ will show inaccurate accuracy due to their brittleness under the existence of adversarial perturbation.

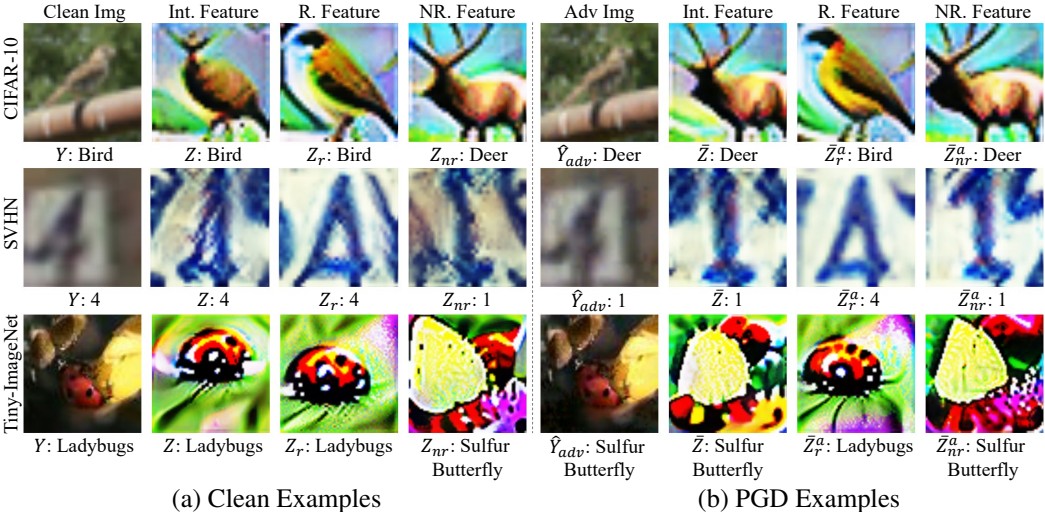

|  | Clean Img | Int. Feature | R. Feature | NR. Feature | Adv Img | Int. Feature | R. Feature | NR. Feature |
|---|---|---|---|---|---|---|---|---|
| CIFAR-10 | $Y$: Bird | $Z$: Bird | $Z_r$: Bird | $Z_{nr}$: Deer | $\hat{Y}_{adv}$: Deer | $\bar{Z}$: Deer | $\bar{Z}_r^a$: Bird | $\bar{Z}_{nr}^a$: Deer |
| SVHN | $Y$: 4 | $Z$: 4 | $Z_r$: 4 | $Z_{nr}$: 1 | $\hat{Y}_{adv}$: 1 | $\bar{Z}$: 1 | $\bar{Z}_r^a$: 4 | $\bar{Z}_{nr}^a$: 1 |
| Tiny-ImageNet | $Y$: Ladybugs | $Z$: Ladybugs | $Z_r$: Ladybugs | $Z_{nr}$: Sulfur Butterfly | $\hat{Y}_{adv}$: Sulfur Butterfly | $\bar{Z}$: Sulfur Butterfly | $\bar{Z}_r^a$: Ladybugs | $\bar{Z}_{nr}^a$: Sulfur Butterfly |

(a) Clean Examples          (b) PGD Examples

Figure 3: Feature visualization [25] for the intermediate feature (Int.), robust feature (R.), and non-robust feature (NR.). The class labels under each image indicate the predicted results of the corresponding features propagated by $f_{l+}(\cdot)$. Note that the visualization of the non-robust features displays semantic similarity of the misclassified classes of the adversarial examples. Please see more visualization results in Appendix E.

In Table 1, we analyze evaluation results of the disentangled features under standard attack algorithms [2, 4, 7] in publicly available datasets [32, 33, 34]. As aforementioned, we apply the robust and non-robust channel index optimized from the clean examples to the adversarial features $\bar{Z}$, and estimate their accuracy (*i.e.,* $f_{l+}(\bar{Z}_r)$ and $f_{l+}(\bar{Z}_{nr})$). As in the table, $\bar{Z}_r$ still shows constant robust accuracy regardless of the adversarial perturbation, even in the high-confidence adversarial attack [4]. On the other hand, we can find that the classification accuracy of $\bar{Z}_{nr}$ steeply degrades as the attacks get stronger, which coincides with the properties of the robust and non-robust features. To further support our experiments, we illustrate the correlation between the disentangled features and true labels, using 2D t-SNE plot [31]. In the case of clean examples as in Fig. 2(a), the robust features $Z_r$ exhibit separable clusters on the target labels, while the non-robust features $Z_{nr}$ show a partially disorganized tendency. When adversarial perturbation [7] exists, the attacked features $\bar{Z}_{nr}$ represents more collapsed t-SNE visualization as shown in Fig. 2(b). Notably, we can observe that $\bar{Z}_r$ still sustain highly clustered results even in the attacked condition.

## 3.2 Feature Visualization of Robust and Non-robust Features

We have identified the existence of the robust and non-robust features using the bottleneck. Then, we wonder about a way of interpreting the semantic information in the feature space with respect to human-perception. Analyzing semantic representation of DNNs is a wide research area to understand their decision [20, 22, 35]. In adversarial settings, several studies [14, 36] argued that a robust classifier has more meaningful (*i.e.,* perceptually-aligned) loss gradients in the input space. Engstrom *et al.* [24] further endeavored to interpret robust feature representation using feature visualization [25, 37, 38]. In this manner, we explore whether the disentangled feature representation from the bottleneck indeed has human-perceptible information in the intermediate feature space.

Feature visualization is an optimization-based method that maximizes specific activation of feature units [38, 39], such that $X' = \operatorname{argmax}_X(a_i^l(X))$, where $a_i^l(\cdot)$ indicates feature activation of $i$-th unit in the $l$-th layer. To understand what conjunction of the robust and non-robust feature units truly interacts with the target labels, we optimize each distilled feature and create their visual explanations. We adopt direct visualization method [25] that has various regularization techniques (*e.g.,* frequency penalization and transformations) to yield better representative visual quality.

We employ $i_r^a$ and $i_{nr}^a$ for the adversarial examples $X_{adv}$, which is obtained by optimizing $\mathcal{L}_{X_{adv}}(\sigma)$ instead of Eq. (4). Then, we define $\bar{Z}_r^a$ and $\bar{Z}_{nr}^a$ as robust and non-robust features of the adversarial examples, satisfying $\bar{Z}_r^a = i_r^a \cdot \bar{Z}$ and $\bar{Z}_{nr}^a = i_{nr}^a \cdot \bar{Z}$. After distilling robust and non-robust features with their corresponding index, we maximize the selected feature unit activation, respectively. The feature visualization results of distilled features are illustrated in Fig. 3.

Table 2: The prediction accuracy of the non-robust features $\hat{Z}_{nr}^a$ for attacked labels $\hat{Y}_{adv}$. The input $\hat{Z}_{nr}^a$ is the non-robust features of the corresponding attack methods. To clearly show the correlation between adversarial examples and the non-robust features, we evaluate the accuracy under the condition of successfully attacked examples (*i.e.*, $Y \neq \hat{Y}_{adv}$).

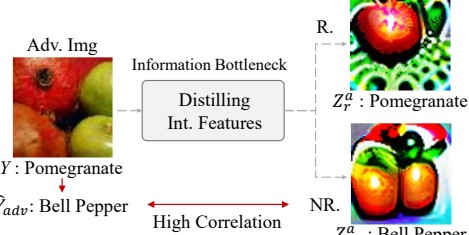

| Attack | CIFAR-10 | | SVHN | | Tiny-ImageNet | |
|---|---|---|---|---|---|---|
| | VGG | WRN | VGG | WRN | VGG | WRN |
| FGSM [2] | 92.78 | 94.35 | 94.90 | 96.13 | 63.39 | 60.82 |
| PGD [7] | 93.43 | 95.06 | 96.21 | 96.44 | 65.84 | 63.64 |
| CW [4] | 93.72 | 94.42 | 95.75 | 97.65 | 55.68 | 56.84 |

Figure 4: An example of highly correlated adversarial prediction with NR. Both $\bar{Y}_{adv}$ and $\bar{Z}_{nr}^a$ output same prediction "Bell Pepper".

As in the figure, we can clearly recognize the semantic information of true labels $Y$ and attacked predictions $\hat{Y}_{adv}$ in the intermediate features ($Z$ and $\bar{Z}$). Interestingly, what we can observe is: (i) the distilled features have semantic information by themselves and maintain their information, even under the adversarial perturbation, (ii) when the adversarial perturbation exists, the brittleness of non-robust features is intensified and reflected onto $\bar{Z}$. Thus, the visualization of $\bar{Z}$ and $\bar{Z}_{nr}^a$ looks similar, and they manipulate the target prediction to same adversarial prediction. Unlike the previous work [15] that has argued the non-robust features solely have incomprehensible property, the visualization of the distilled features from our bottleneck represent recognizable outputs even for the adversarial examples, and provides a decisive key to interpret the cause of adversarial examples.

## 3.3 Adversarial Prediction is Highly Correlated with Non-robust Features

We have observed that the non-robust features optimized from the bottleneck are brittle and easily manipulated under the adversarial perturbation, while robust features maintain substantial prediction results for the target labels. Then, if the primary cause of the adversarial examples indeed belongs to non-robust features, it is natural to examine the correlation between the classification outputs of the non-robust features and the adversarial prediction induced by adversarial attacks. Accordingly, we identically apply our IB loss on the adversarial examples $X_{adv}$ [2, 7, 4], and find their corresponding robust and non-robust channel index using Eq. (5).

To enlighten the correlation of non-robust features and the adversarial prediction, we evaluate the model prediction of $\bar{Z}_{nr}^a$ for the attacked labels $\hat{Y}_{adv}$ that can be written as follows: $\hat{Y}_{nr}^a = f_{l+}(\bar{Z}_{nr}^a)$. We set the condition of $\hat{Y}_{adv}$ as successfully attacked labels (*i.e.*, $Y \neq f(X_{adv})$) to definitely show the relationship between the prediction $\hat{Y}_{nr}^a$ of non-robust features and the adversarial prediction $\hat{Y}_{adv}$ of adversarial examples. In Table 2, we summarize the accuracy of $\hat{Y}_{nr}^a$ for the successfully attacked label $\hat{Y}_{adv}$ in standard attack methods. Generally, we can observe that the non-robust features are highly predictive on $\hat{Y}_{adv}$ in standard low dimensional datasets such as CIFAR-10 and SVHN. Even in a large dataset (*i.e.,* Tiny-ImageNet), the non-robust features are remarkably correlated with the attacked prediction. A brief explanation of the highly correlated example is described in Fig. 4.

## 3.4 Bottleneck Controls Information Flow of Robust and Non-robust Features

In this analysis, we will investigate how the bottleneck affects the information flow of the robust and non-robust features and clarify their relation. Recall that the bottleneck refines informative features from the given image samples, and $\beta$ regulates the total amount of the information that flows into $Z$. We will compare classification accuracy for the robust and non-robust features along $\beta$ value and analyze the changes of information flow assigned to each disentangled feature.

In Fig. 5, as $\beta$ value increases, the accuracy of the robust features is getting higher and decreases after a specific threshold. As theoretically mentioned in 2.1, we can infer that a suitable choice of the $\beta$ can filter out robust feature units in the intermediate layer. For the excessive $\beta$ value, we can observe that the bottleneck cannot accurately disentangle adversarial features. For example, the accuracy of the robust and non-robust features are reversed after $\beta = 5.0$ in the particular networks of CIFAR-10 and SVHN datasets. In addition, as in Fig. 5(a) and (b), the accuracy of the non-robust features progressively increases. Such results indicate a few robust feature units that are not distilled

Table 3: Comparing attack performance for FGSM [2], BIM [40], PGD [7], CW [4], AutoAttack (AA) [41], FAB [42], and non-robust feature attack denoted by NRF. We adversarially train VGG-16 and WRN-28-10 on $L_\infty$ norm $\gamma = 0.03$ perturbation for CIFAR-10, SVHN, and Tiny-ImageNet with PGD adversarial training [7] (ADV) and advanced defense methods: TRADES [43] and MART [44].

| Dataset | Method | VGG-16 | | | | | | | | WRN-28-10 | | | | | | | |
|---|---|---|---|---|---|---|---|---|---|---|---|---|---|---|---|---|---|
| | | Clean | FGSM | BIM | PGD | CW | AA | FAB | NRF | Clean | FGSM | BIM | PGD | CW | AA | FAB | NRF |
| CIFAR-10 | ADV | 79.7 | 51.3 | 46.5 | 44.7 | 40.3 | 42.0 | 40.9 | **27.4** | 82.6 | 56.4 | 52.8 | 51.6 | 45.5 | 49.8 | 49.0 | **17.1** |
| | TRADES | 78.2 | 54.5 | 51.7 | 50.9 | 43.0 | 49.5 | 46.3 | **31.2** | 83.0 | 57.9 | 55.0 | 53.9 | 46.7 | 52.4 | 49.8 | **26.8** |
| | MART | 73.5 | 54.2 | 52.2 | 51.7 | 42.2 | 50.6 | 45.1 | **31.4** | 83.4 | 59.0 | 56.0 | 54.7 | 46.5 | 52.8 | 50.2 | **19.6** |
| SVHN | ADV | 90.4 | 63.7 | 52.1 | 48.8 | 33.3 | 39.9 | 41.6 | **12.6** | 93.5 | 73.9 | 64.8 | 61.1 | 40.7 | 55.5 | 56.6 | **13.4** |
| | TRADES | 90.4 | 65.3 | 59.0 | 57.0 | 44.8 | 53.5 | 50.0 | **14.3** | 93.9 | 72.9 | 63.9 | 60.4 | 42.0 | 55.0 | 55.4 | **10.1** |
| | MART | 90.5 | 65.1 | 59.7 | 57.8 | 46.4 | 53.0 | 47.0 | **16.1** | 94.1 | 73.0 | 64.5 | 61.1 | 42.3 | 55.4 | 56.0 | **8.1** |
| Tiny-ImageNet | ADV | 34.0 | 17.5 | 16.5 | 16.1 | 12.0 | 15.4 | 12.2 | **6.7** | 43.1 | 20.4 | 19.3 | 18.8 | 13.5 | 18.1 | 14.2 | **5.3** |
| | TRADES | 38.7 | 20.1 | 19.1 | 18.7 | 13.9 | 17.8 | 13.3 | **7.8** | 47.2 | 26.7 | 25.6 | 25.2 | 17.4 | 24.4 | 17.7 | **9.6** |
| | MART | 38.4 | 20.6 | 19.5 | 19.1 | 14.1 | 18.3 | 14.7 | **9.2** | 48.5 | 27.4 | 26.1 | 25.7 | 17.5 | 25.0 | 17.8 | **9.9** |

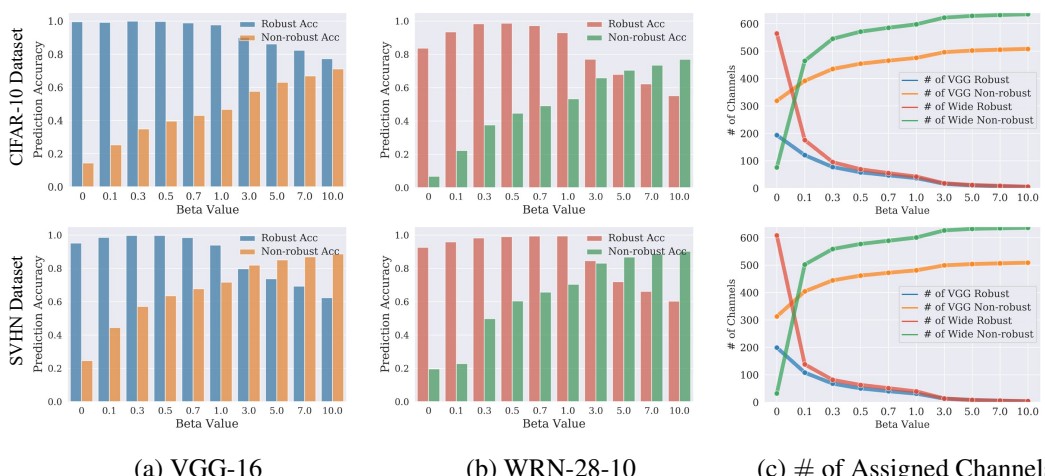

(a) VGG-16      (b) WRN-28-10      (c) # of Assigned Channels

Figure 5: The accuracy of robust and non-robust features along $\beta$ value, and the number of assigned channels in the $l$-th feature representation. Note that each color of the lines in (c) corresponds to the same color in the bar plots of (a) and (b). The total number of channels is equivalent to the size of $C$.

from IB are gradually flowing into the non-robust side and dichotomized as non-robust units, thus producing more higher accuracy. It coincides with the analysis of the number of the assigned robust and non-robust channels in Fig. 5(c). We can observe that the number of robust channels constantly diminishes, whereas that of the non-robust channels increases. It indicates the bottleneck delicately weighs the information flow that should be precisely assigned to the robust or non-robust units.

From our analysis section, in conclusion, we have demonstrated the existence of the distilled features using the bottleneck: the robust and non-robust features in the intermediate feature representation. In addition, we have revealed that easily manipulated property of non-robust features is the primary cause of adversarial examples through their high correlation with the adversarial prediction. Based on the fact that the non-robust features break the model prediction, we will suggest an effective way of enhancing the adversarial attack, utilizing the gradient of non-robust features in section 4.

## 4 Amplifying Brittleness of Non-Robust Features for Attack

In this section, we intentionally increase the non-robust noise variation $\sigma_{nr}$ to make model classifier fooled based on Eq. (6). Note that increasing $\sigma_{nr}$ induces high erroneous term $\Delta$, thereby going to deviate from the robust prediction. However, this variation $\sigma_{nr}$ is merely an optimizing parameter that cannot be controlled manually. Thus, in order to secondarily have the effect of enlarging $\sigma_{nr}$, we alternatively utilize the gradients of the non-robust features directly connected with the model

Table 4: Comparison of classification accuracy for adversarial examples generated by maximizing (↑) or minimizing (↓) the magnitude of robust ($\mathcal{G}_r$) or non-robust feature gradients ($\mathcal{G}_{nr}$).

| Dataset | Method | VGG-16 | | | | | WRN-28-10 | | | | |
|---|---|---|---|---|---|---|---|---|---|---|---|
| | | Clean | $\|\mathcal{G}_{nr}\|_2 \uparrow$ | $\|\mathcal{G}_{nr}\|_2 \downarrow$ | $\|\mathcal{G}_r\|_2 \uparrow$ | $\|\mathcal{G}_r\|_2 \downarrow$ | Clean | $\|\mathcal{G}_{nr}\|_2 \uparrow$ | $\|\mathcal{G}_{nr}\|_2 \downarrow$ | $\|\mathcal{G}_r\|_2 \uparrow$ | $\|\mathcal{G}_r\|_2 \downarrow$ |
| | ADV | 79.7 | **27.4** | 67.5 | **35.9** | 74.6 | 82.6 | **17.1** | 74.8 | **28.9** | 79.5 |
| CIFAR-10 | TRADES | 78.2 | **31.2** | 71.6 | **38.2** | 77.1 | 83.0 | **26.8** | 73.9 | **30.3** | 79.9 |
| | MART | 73.5 | **31.4** | 63.8 | **39.6** | 69.1 | 83.4 | **19.6** | 74.3 | **24.5** | 79.4 |
| | ADV | 90.4 | **12.6** | 71.9 | **20.8** | 71.5 | 93.5 | **13.4** | 88.0 | **15.6** | 93.4 |
| SVHN | TRADES | 90.4 | **14.3** | 68.4 | **27.3** | 82.3 | 93.9 | **10.1** | 88.1 | **13.2** | 93.3 |
| | MART | 90.4 | **16.1** | 66.2 | **31.6** | 84.9 | 94.1 | **8.1** | 87.7 | **9.0** | 91.5 |
| | ADV | 34.0 | **6.7** | 26.1 | **9.7** | 29.0 | 43.1 | **5.3** | 38.9 | **15.5** | 39.9 |
| Tiny-ImageNet | TRADES | 38.7 | **7.8** | 30.7 | **11.8** | 33.8 | 47.2 | **9.6** | 40.9 | **16.9** | 44.4 |
| | MART | 38.4 | **9.2** | 29.0 | **13.4** | 32.5 | 48.5 | **9.9** | 41.3 | **17.2** | 45.7 |

prediction. The gradients of non-robust features in adversarial examples can be described as follows:

$$\mathcal{G}_{nr} = \frac{\partial}{\partial \bar{Z}_{nr}} \mathcal{L}_{base}(f(X + \delta), Y), \tag{7}$$

where we define a baseline loss as $\mathcal{L}_{base}(f(X), Y) = \|\delta\|_2 + c \cdot \max(\max_{i \neq Y}(f(X)_i) - f(X)_i, 0)$, instead of cross-entropy loss due to its empirical effectiveness of attack performance [4]. In addition, we use a technique of *changes of variables* [4] from $\delta$ to $w$ for generating an imperceptible yet powerful perturbation, such that $\delta = \frac{1}{2}(\tanh(w) + 1) - X$. It serves to smooth out projected gradient descent that clips prematurely to prevent adversarial examples falling into the extreme image domain.

To compute the gradient practically, we firstly calculate $\frac{\partial}{\partial \bar{Z}} \mathcal{L}_{base}$, and multiply it to $\frac{\partial}{\partial \bar{Z}_{nr}} \bar{Z}$ by chain rule. Here, the latter gradient equals to non-robust channel index $i_{nr}$, because the intermediate features of adversarial examples can be re-written as: $\bar{Z} = i_r \cdot \bar{Z}_r + i_{nr} \cdot \bar{Z}_{nr}$, and the derivative of $\bar{Z}$ over $\bar{Z}_{nr}$ equals to $i_{nr}$. Thus, the gradients of the non-robust features $\mathcal{G}_{nr}$ can be simplified as $i_{nr} \cdot \frac{\partial}{\partial \bar{Z}} \mathcal{L}_{base}$. Using the gradient, we suggest an attack to non-robust features (NRF) by optimizing the following objective:

$$\min_{\delta} \mathcal{L}_{base}(f(X + \delta), Y) - \left\| i_{nr} \cdot \frac{\partial}{\partial \bar{Z}} \mathcal{L}_{base} \right\|_2. \tag{8}$$

In Table 3, NRF shows more effective attack performance than the other standard adversarial attacks in [7, 43, 44], since NRF strengthens the gradients of non-robust features that contains the same effect of increasing $\sigma_{nr}$ to disturb accurate model prediction.

Moreover, we conduct an ablation study on the gradients of robust and non-robust features to probe their influence of prediction changes in Table 4. As expected, maximizing $\mathcal{G}_{nr}$ shows more effective attack performance than minimizing it. It is because maximizing $\mathcal{G}_{nr}$ has an alternative effect of increasing the non-robust noise variation $\sigma_{nr}$ (*i.e.,* large erroneous term $\Delta \uparrow$) in Eq. (6). Whereas, manipulating the gradients of robust features $\mathcal{G}_r$ shows less attack performance than controlling $\mathcal{G}_{nr}$, since it cannot directly handle brittle features in the intermediate feature representation. Especially, it seems difficult to break model prediction, while weakening the gradient of robust features, whose noise variation $\sigma_r$ have invariance of target prediction even under the adversarial perturbation.

## 5 Related Work

Various works have been tried to figure out the reason for adversarial vulnerability in the intermediate feature representation. Inkawhich *et al.* [45] analyzed how the intermediate features are changed by adversarial attacks and measured layer- and class-wise feature distributions. Engstrom *et al.* [24] pointed out that there is a shortcoming of DNNs and their embedding, that is, the primary features used in DNNs are contrasting with what human uses. Also, they argued that the robust optimization to learn robust features could address this shortcoming by encoding high-level representations of input data. Jacobsen *et al.* [46] argued that the reason for adversarial vulnerability lies in invariant characteristics of DNNs to task-relevant features. This invariance makes most regions of input space brittle to adversarial attacks so that the classifiers become relying on a few highly predictive features. In addition, recent works [14, 15] have suggested that the existence of the vulnerability is on the non-robust features, which are inherently included in the data and have unrecognizable properties.

Our work is in line with the concept of the non-robust feature. However, unlike the aforementioned works that directly generated the robust and non-robust datasets to analyze their properties, we reveal that the robust and non-robust features can be completely disentangled in the feature space using Information Bottleneck, and they have semantic information by themselves in fact.

## 6    Conclusion and Discussion

**Conclusion**. To understand where the adversarial brittleness comes from, we have investigated information flow in the intermediate feature space, using Information Bottleneck. By estimating the feature importance of each feature unit based on the added noise variation, we introduce a novel method to explicitly distill the robust and non-robust features in the feature representation. Through extensive analysis of the distilled features, we figure out how the feature units interact with target labels, and corroborate that the non-robust features are highly correlated with the adversarial prediction. In addition, we directly visualize the distilled features in the intermediate feature space and reveal that they have recognizable semantic information by themselves. Based on the properties of the distilled features, we suggest a non-robust feature attack (NRF) utilizing the brittleness in the feature space.

**Discussion**. The adversarial examples, including our work, can potentially cause negative impacts on various machine learning applications, such as autonomous driving [47] and medical image process [48]. However, by analyzing the nature of robust and non-robust features that can be a primary cause of the adversarial examples, our work contributes important societal impacts in this research field. In this paper, we propose a way of utilizing the gradient of brittle features in the intermediate feature representation (NRF). In future works, we hope to bridge the gap of deploying distilled features into diverse applications such as adversarial detection and defense strategies. Moreover, we would like to note that the last convolutional layer is set to a distilling criterion for Information Bottleneck, since the high-level concepts are included in the higher layer of DNNs. In this sense, it seems a reasonable choice, but how the robust or non-robust features in the lower dimension interact with the much higher dimensional feature units can be another undisclosed future direction to investigate.

## Acknowledgments and Disclosure of Funding

This work was conducted by Center for Applied Research in Artificial Intelligence (CARAI) grant funded by DAPA and ADD (UD190031RD).

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
