# A Implementation Details

**Datasets.** We conduct exhaustive experiments on not only a standard low dimensional dataset on $32\times32$ pixels with 10 classes: CIFAR-10 [1] and SVHN [2], but also more challenging datasets Tiny-ImageNet [3]. Tiny-ImageNet is a small subset of ImageNet dataset, containing 100,000 training images, 10,000 validation images, and 10,000 testing images separated in 200 different classes, dimensions of which are $64\times64$ pixels.

**Baseline Models.** We adopt two baseline models: VGG-16 and WideResNet-28-10 trained with $L_\infty$ perturbation magnitude $\gamma = 0.03$ as a standard PGD training [4]. For the models' adversarial training, we use learning rate of 0.01 with SGD [5] in 60 epoch for early stopping [6] to reduce overfitting that in fact harms adversarial robustness. In addition, we take a step scheduler to lower the learning rate by 0.2 times every 20 epoch, and we hold 0.01 parameter of weight decay. For PGD settings, we set the number of steps $N$ to 10 for gradient clipping, and we set the step size $\eta = \frac{2.3}{N} \times \gamma$ to $\frac{2.3}{10} \times 0.03$.

**Our Information Bottleneck.** Below algorithm represents how robust and non-robust features are distilled by our Information Bottleneck (IB). We choose IB's coefficient $\beta$ to 0.3 for VGG-16 on CIFAR-10 and SVHN, and 0.5 for WideResNet-28-10 on CIFAR-10, and 1.0 for WideResNet-28-10 on SVHN. These selections are determined from Fig. 5 in our manuscript. For Tiny-ImageNet, we set $\beta = 0.3$ both for VGG-16 and WideResNet-28-10. Note that we use IB's learning rate $\alpha$ of 0.1 with Adam [7] in 200 iterations. In practical, we consider noise variation $\sigma$ ($\in \mathbb{R}^C$) as optimizing parameter, but it may be learned to be non-positive values. To address it, we apply SoftPlus to the noise variation to get positive values. Here, SoftPlus is a function of $\log(1 + \exp(\cdot))$.

---

**Algorithm 1**

---

**Require:** $X, Y$ : an image and its target label, $\sigma^-$ : (pseudo) noise variation ($\sigma^- \in \mathbb{R}^C$).
1: $Z \leftarrow f_l(X), (Z \in \mathbb{R}^{C \times H \times W})$         ▷ Extracting an intermediate feature
2: $\sigma_z^2 \leftarrow Var(Z), (\sigma_z^2 \in \mathbb{R}^C)$         ▷ Calculating (channel-wise) feature variation
3: $\sigma^- \leftarrow 0 \in \mathbb{R}^C$         ▷ Initialize the noise variation to zero
4: **for** iteration **do**         ▷ Our IB procedure
5:      $\sigma \leftarrow \log(1 + \exp(\sigma^-))$         ▷ SoftPlus to be positive values
6:      $Z_I \leftarrow Z + \sigma \cdot \epsilon, (\epsilon \sim \mathcal{N}(0, I))$         ▷ Sampling an informative feature
7:      $\hat{Y} \leftarrow f_{l+}(Z_I)$         ▷ Model Prediction
8:      $\mathcal{L}_{CE} \leftarrow -\sum_{k=1}^{C} Y_k \log \hat{Y}_k$         ▷ Cross-entropy loss
9:      $\mathcal{L}_I \leftarrow \frac{1}{2} \sum_{k=1}^{C} [\frac{\sigma_{z_k}^2}{\sigma_k^2} + \log \frac{\sigma_k^2}{\sigma_{z_k}^2} - 1]$         ▷ Information loss
10:     $\mathcal{L}_X(\sigma^-) \leftarrow \mathcal{L}_{CE} + \beta \mathcal{L}_I$         ▷ Our IB loss
11:     $\sigma^- \leftarrow \sigma^- - \alpha \frac{\partial}{\partial \sigma^-} \mathcal{L}_X(\sigma^-)$         ▷ Update noise variation
12: **end for**
13: $\sigma \leftarrow \log(1 + \exp(\sigma^-))$         ▷ SoftPlus to be positive values
14: $T = max(\sigma_z^2)$         ▷ Obtaining a maximum tolerance
15: $i_r \leftarrow \mathbb{1}(\sigma^2 > T), (i_r \in \mathbb{R}^C)$         ▷ Finding a robust channel index
16: $i_{nr} \leftarrow 1 - i_r, (i_{nr} \in \mathbb{R}^C)$         ▷ Finding a non-robust channel index
17: $Z_r \leftarrow i_r \cdot Z$         ▷ Distilling a robust feature
18: $Z_{nr} \leftarrow i_{nr} \cdot Z$         ▷ Distilling a non-robust feature

---

**Validation.** The main purpose of this work is to demonstrate the impacts of robust and non-robust features to model prediction. In order to measure classification accuracy, we take various adversarial attacks of FGSM [8], BIM [9], AutoAttack [10], and FAB [10, 11] under $L_\infty$ perturbation magnitude $\gamma = 0.03$. In particular, we set hyperparameters of BIM that step number $N$ is automatically arranged on $\gamma = 0.03$ neighborhood, and step size $\eta = 1/255$. Besides, hyperparameters of AutoAttack and FAB are chosen that $N = 100$, $\rho = 0.75$, $\alpha_{max} = 0.1$, $\eta = 1.05$, $\beta = 0.9$ of which notations are used in their papers. In addition, we also use a strong attack of CW [12] with constraining $L_2$ distance metric of 0.01, which overcomes defensive distillation [13], encompassing a range of attacks cast through the same optimization framework with Lagrangian relaxation. Note that the hyperparameters of CW equal to that $c = 0.1$, $\kappa = 0$, $N = 200$, learning rate= 0.1 with Adam. For NRF attack in Section 4 of our manuscript, we equally use those of CW with $L_2$ distance metric of 0.01 to fairly validate its effectiveness for intensifying brittleness of non-robust features.

# B Deep Information Bottleneck

Let $X$ denotes inputs and $Y$ denotes (one-hot encoded) target labels corresponding to the input, and $Z$ indicates the intermediate features of deep neural networks (DNNs). The following equation represents an objective of Information Bottleneck (IB).

$$\max_{Z} I(Z, Y) - \beta I(Z, X),\qquad(1)$$

where $I$ denotes mutual information, and $\beta$ represents the degree of restraining input information $X$. Here, the first term can be expressed as follows:

$$I(Z, Y) = \int p(y, z) \log \frac{p(y, z)}{p(y)p(z)} dy dz$$

$$= \int p(y, z) \log \frac{p(y \mid z)}{p(y)} dy dz,\qquad(2)$$

where a true feature probability $p(z)$ is erased on the fraction. To apply it to DNNs, model classifier denoted by $q(y \mid z)$ is introduced to perform model prediction. Based on variational inference [14], it is a closed-form for a true likelihood $p(y \mid z)$. This approximation is formulated by KL divergence (always positive) and then the following inequality is constructed as follows:

$$D_{KL}[p(Y \mid Z) \mid\mid q(Y \mid Z)] \geq 0 \quad \Rightarrow \quad \int p(y \mid z) \log p(y \mid z) dy \geq \int p(y \mid z) \log q(y \mid z) dy \qquad(3)$$

which helps getting IB's objective to be tractable. With this equality, the first term in Eq. (1) can be represented to a lower bound which can be written as:

$$I(Z, Y) \geq \int p(y, z) \log \frac{q(y \mid z)}{p(y)} dy dz$$

$$= \int p(y, z) \log q(y \mid z) dy dz - \int p(y, z) \log p(y) dy dz$$

$$= \int p(y, z) \log q(y \mid z) dy dz - \int p(y) \log p(y) dy \qquad(4)$$

$$= \int p(y, z) \log q(y \mid z) dy dz + H(Y)$$

$$\geq \int p(y, z) \log q(y \mid z) dy dz,$$

where a positive constant $H(Y) = - \int p(y) \log p(y) dy \geq 0$ denotes *Shannon entropy* of target labels (ignored). Next, the second term in Eq. (1) is described as:

$$I(Z, X) = \int p(z, x) \log \frac{p(z, x)}{p(x)p(z)} dz dx$$

$$= \int p(z, x) \log \frac{p(z \mid x)}{p(z)} dz dx,\qquad(5)$$

where a dataset probability $p(x)$ is erased on the fraction. Here, an approximate feature probability $q(Z)$ is introduced to approximate the true feature probability $p(Z)$. As similar with Eq. (3), the relationship between $q(Z)$ and $p(Z)$ can be written and then it builds the following equality:

$$D_{KL}[p(Z) \mid\mid q(Z)] \geq 0 \quad \Rightarrow \quad \int p(z) \log p(z) dz \geq \int p(z) \log q(z) dz \qquad (6)$$

By using it, the second term is constructed with a upper bound as follows:

$$I(Z, X) \leq \int p(z, x) \log \frac{p(z \mid x)}{q(z)} dz dx$$

$$= \int p(x) p(z \mid x) \log \frac{p(z \mid x)}{q(z)} dz dx \qquad (7)$$

$$= \mathop{\mathbb{E}}_{X \sim p(X)} \left[ D_{KL}[p(Z \mid X) \mid\mid q(Z)] \right],$$

where a feature likelihood is denoted by $p(z \mid x)$. To sum it up, the IB's objective can be re-formulated with a lower bound as follows:

$$I(Z, Y) - \beta I(Z, X) \geq \int p(y, z) \log q(y \mid z) dy dz - \beta \underbrace{\mathop{\mathbb{E}}_{X \sim p(X)} [D_{KL}[p(Z \mid X) \mid\mid q(Z)]]}_{\mathcal{L}_I}, \qquad (8)$$

where $\mathcal{L}_I$ denotes a information loss computed by KL divergence [15] between the feature likelihood $p(Z \mid X)$ and the approximate feature probability $q(Z)$. Without the true feature probability $p(Z)$, employing the lower bound is an alternative way of maximizing the IB's objective so that IB is easily applied to DNNs to make the intermediate features $Z$ become rich information. In practical, the first term in the lower bound is simplified to the expected log-likelihood in a form of cross entropy $\mathcal{L}_{CE}$, which can be written as follows:

$$\int p(y, z) \log q(y \mid z) dy dz = \int p(z) p(y \mid z) \log q(y \mid z) dy dz$$

$$= \int p(z, x) p(y \mid z) \log q(y \mid z) dy dz dx$$

$$= \int p(x) p(z \mid x) p(y \mid z) \log q(y \mid z) dy dz dx \qquad (9)$$

$$= \mathop{\mathbb{E}}_{X \sim p(X), Z \sim p(Z \mid X)} \left[ \int p(y \mid z) \log q(y \mid z) dy \right]$$

$$= \mathop{\mathbb{E}}_{X \sim p(X), Z \sim p(Z \mid X)} [-\mathcal{L}_{CE}],$$

where $p(y \mid z)$ indicates the true likelihood considered as a target label $y$ corresponding $z$. Note that since we consider a non-random model in the inference phase, the expected log-likelihood can be calculated to $\mathop{\mathbb{E}}_{X \sim p(X)} [-\mathcal{L}_{CE}]$ without repeatedly sampling the intermediate feature $Z$ given an input $X$. Hence, IB's objective is re-written as follows:

$$\min_{Z} \mathcal{L}_{CE} + \beta \mathcal{L}_I. \qquad (10)$$

# C  Model Prediction for Our IB by Taylor Expansion

We analyze impacts of robust $Z_r = i_r \cdot f_l(X)$ and non-robust features $Z_{nr} = i_{nr} \cdot f_l(X)$ to model robustness by expanding model prediction of informative features $Z_I = f_l(X) + \sigma \cdot \epsilon$ to Taylor approximation with its convergence of local minima [16, 17] as follows:

$$f_{l+}(f_l(X) + \sigma \cdot \epsilon) = f_{l+}(f_l(X) + \sigma_r \cdot \epsilon) + \underbrace{\left[ \frac{\partial}{\partial \sigma_r} f_{l+}(f_l(X) + \sigma_r \cdot \epsilon) \right]^T \sigma_{nr}}_{\Delta}, \qquad (11)$$

where robust noise variation $\sigma_r = i_r \cdot \sigma$ and non-robust noise variation $\sigma_{nr} = i_{nr} \cdot \sigma$. Now, we will prove the above formulation.

***Proof.*** Taylor approximation represents a number as a polynomial that has a very similar value to the number in a neighborhood around a specified value. It is a powerful tool to approximate a function that can be intractable. It is evaluated as infinite sums and integrals of the function's derivatives at a single point. Basically, it can be written with an arbitrary function $F : \mathbb{R} \to \mathbb{R}$ as follows:

$$F(a + \Delta a) = F(a) + \sum_{k=1}^{\infty} \frac{(\Delta a)^k}{k!} \frac{\partial^k}{\partial a^k} F(a). \qquad (12)$$

Once the difference $\Delta a \in \mathbb{R}$ gets small enough, it can be expressed as a first-order polynomial as:

$$F(a + \Delta a) = F(a) + \Delta a \frac{\partial}{\partial a} F(a). \qquad (13)$$

Here, we extend it to deal with a multi-variable function $\mathcal{F} : \mathbb{R}^M \to \mathbb{R}^N$ given small $\Delta a$ as follows:

$$\mathcal{F}(a + \Delta a) = \mathcal{F}(a) + \left[ \frac{\partial}{\partial a} \mathcal{F}(a) \right]^T \Delta a. \qquad (14)$$

Then, dimension of $\mathcal{F}(a)$ and $\frac{\partial}{\partial a}\mathcal{F}(a)$ is each $\mathbb{R}^N$ and $\mathbb{R}^{M \times N}$, such that $a \in \mathbb{R}^M$. In our manuscript, since the non-robust noise variation $\sigma_{nr}$ is small enough, therefore, the model prediction for our IB can be formulated with Taylor expansion of Eq. (14) as follows:

$$f_{l+}(f_l(X) + \sigma \cdot \epsilon) = f_{l+}(f_l(X) + \sigma \cdot (i_r + i_{nr}) \cdot \epsilon)$$

$$= f_{l+}(f_l(X) + (\sigma_r + \sigma_{nr}) \cdot \epsilon)$$

$$= \underbrace{f_{l+}}_{\mathcal{F}} \underbrace{(f_l(X) + \sigma_r \cdot \epsilon}_{a} + \underbrace{\sigma_{nr} \cdot \epsilon}_{\Delta a})$$

$$= \underbrace{f_{l+}}_{\mathcal{F}} \underbrace{(f_l(X) + \sigma_r \cdot \epsilon)}_{a} + \left[ \frac{\partial}{\partial (\underbrace{f_l(X) + \sigma_r \cdot \epsilon}_{a})} f_{l+} \underbrace{(f_l(X) + \sigma_r \cdot \epsilon)}_{a} \right]^T \underbrace{\sigma_{nr} \cdot \epsilon}_{\Delta a}$$

$$= f_{l+}(f_l(X) + \sigma_r \cdot \epsilon) + \left[ \frac{\partial}{\partial \sigma_r} f_{l+}(f_l(X) + \sigma_r \cdot \epsilon) \frac{\partial \sigma_r}{\partial (f_l(X) + \sigma_r \cdot \epsilon)} \right]^T \sigma_{nr} \cdot \epsilon$$

$$= f_{l+}(f_l(X) + \sigma_r \cdot \epsilon) + \left[ \frac{\partial}{\partial \sigma_r} f_{l+}(f_l(X) + \sigma_r \cdot \epsilon) \cdot \frac{1}{\epsilon} \right]^T \sigma_{nr} \cdot \epsilon$$

$$= f_{l+}(f_l(X) + \sigma_r \cdot \epsilon) + \left[ \frac{\partial}{\partial \sigma_r} f_{l+}(f_l(X) + \sigma_r \cdot \epsilon) \right]^T \sigma_{nr}.$$

$$(15)$$

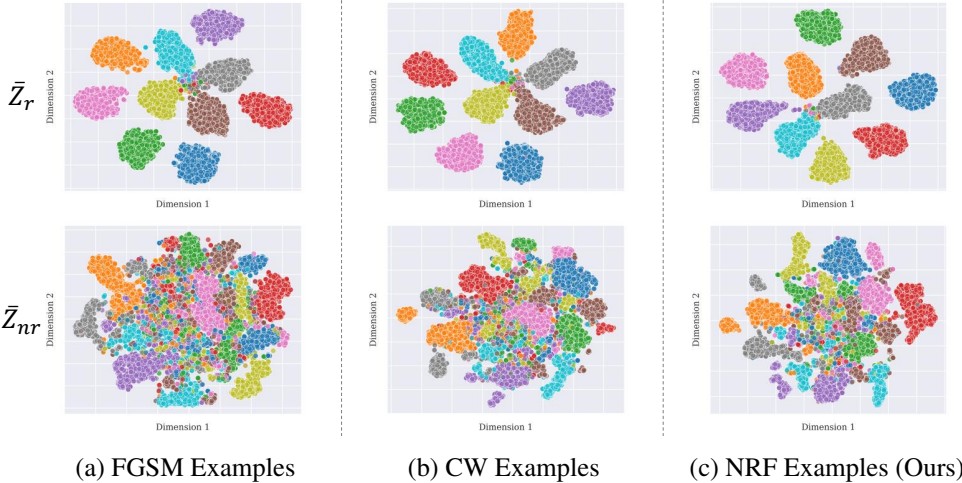

$\bar{Z}_r$

$\bar{Z}_{nr}$

(a) FGSM Examples        (b) CW Examples        (c) NRF Examples (Ours)

Figure 1: Additional t-SNE plot [18] results of FGSM [8], CW [12], and NRF attack in VGG-16. Each cluster indicates high-dimensional distributions of feature representation for 10 object labels in CIFAR-10.

# D   Additional t-SNE Plot Results

In Section 3.1, we have analyzed the properties of the robust and non-robust features in the feature representation. We extend analysis of t-SNE plots to the various attack scenarios for further generalization. The additional results are illustrated in Fig. 1. As in the figure, we can observe that the intermediate robust feature representation $\bar{Z}_r$ of FGSM, CW, and NRF attacks exhibit well clustered tendency along target labels. Whereas, the embedding space of $\bar{Z}_{nr}$ generally shows disorganized clusters due to their brittleness.

# E   Omitted Visual Explanations

**Additional Feature Visualization.** We provide additional feature visualization under various adversarial attack methods including NRF in Fig. 3-5 (CIFAR-10, SVHN, and Tiny-ImageNet are utilized). As in the figure, we can realize that the semantic information in the distilled feature space and manipulated intermediate feature visualization when an attack comes in. Moreover, the distilled features still include the robust and brittle information even in the failed attack examples. Notably, we can infer that the prediction of non-robust features significantly affect to the adversarial prediction, because the characteristic properties of the brittle information reflects to the intermediate features. Through the comprehensive qualitative results, we corroborate the robust and non-robust features indeed have human-perceptible properties and provide an interpretable cause of adversarial examples.

**Visual Interpolation of Distilled Features.** In the previous work, Tsipras *et al.* [19] show linear interpolating results between the clean and PGD-attacked examples in the image domain with large-$\epsilon$, in order to explain perceptually plausible interpolation changes of cross-class relation. As a further work, we conduct an experiment of linear interpolation results, not in the image domain but in the distilled feature space to analyze how small imperceptible changes in the given images affects intermediate feature representation. Through this analysis by showing visual interpolation between the robust and non-robust features, we want to show smooth changes of the visual results towards the adversarial prediction.

In Fig. 6, we illustrate the linear interpolation results of feature visualization [20] between the robust and non-robust features. Each visualization of interpolated feature representation can be written as follows: $Z_{int} = \lambda \cdot Z_r + (1 - \lambda) \cdot Z_{nr}$, where $\lambda$ is interpolation parameter between the robust and non-robust features. In the figure, the first and second row indicates the feature visualization of $Z_{int}$ for the clean example and PGD-attacked examples, respectively. As the proportion of non-robust features in $Z_{int}$ increases, we can observe the visualization smoothly changes from the target label to the adversarial prediction (*e.g.,* deer→bird) in the both examples. Through interpolation results, we

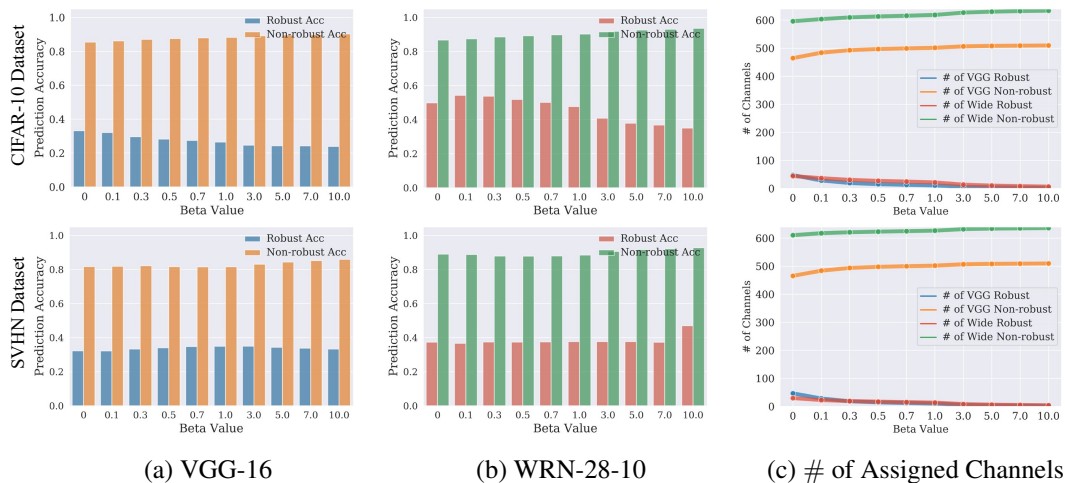

Figure 2: The classification accuracy of robust and non-robust features in standard setting. Along $\beta$ value, and the number of assigned channels in the $l$-th feature representation. Note that each color of the lines in (c) corresponds to the same color in the bar plots of (a) and (b). The total number of channels is equivalent to the size of $C$.

can infer that imperceptible adversarial noise indeed affects to the intermediate feature space and manipulate the model prediction to the adversarial prediction.

# F  Analysis of Information Bottleneck in Standard Setting

In optimizing our bottleneck objective, we have utilized adversarially trained model $f$ to distill the robust and non-robust features. In several works [19, 21], it has been argued that the distinction between robust and non-robust features are established in adversarial setting, since the non-robust features are not solely brittle, but highly predictive concurrently in standard setting. In the nature of standard training paradigm, any predictive features including non-robust features can be utilized to estimate target labels, thus disentangling feature representation based on the predictivity cannot be an optimal criterion. To verify such properties in the bottleneck, we conduct same experiments as in Section 3.4, but in standard setting.

In Fig. 2, the bottleneck cannot disentangle the robust and non-robust features accurately along $\beta$ changes. As we can find, the classification accuracy of using the non-robust features are more higher than those of the robust features, which is opposite results of adversarially trained model analyzed in Section 3.4. Moreover, the number of channels assigned to robust features are significantly small amount than the non-robust features, since the non-robust features play a dominant role in canonical standard classification setting [21]. Accordingly, we utilize the adversarially trained model as $f$, which is trained to learn robust representation [22] and not relying on the non-robust features to predict target labels.

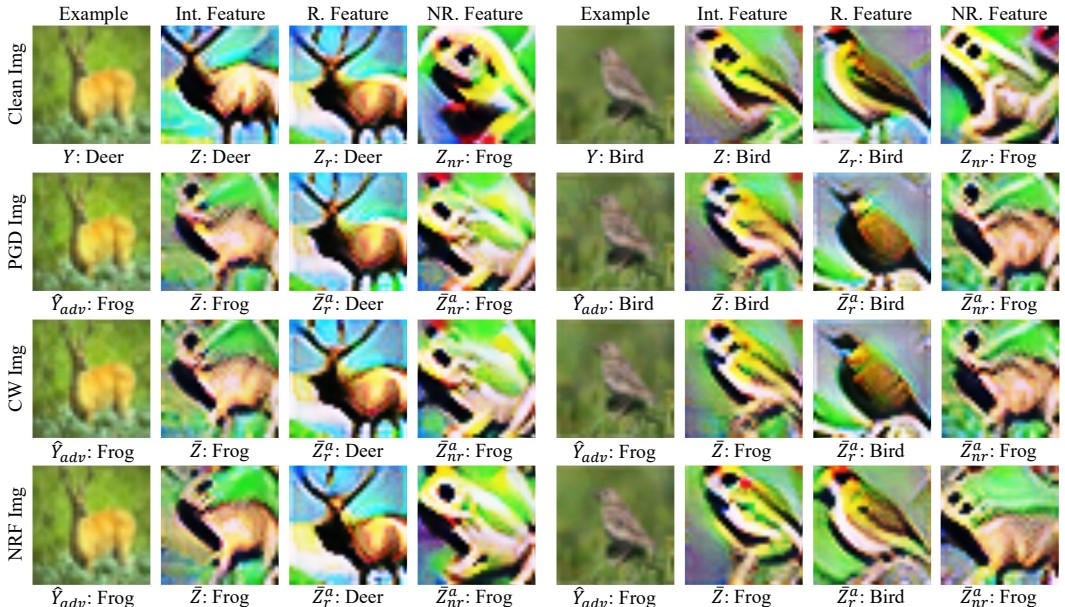

Figure 3: Feature visualization of CIFAR-10. The class labels under each image indicate the target prediction results of the corresponding features.

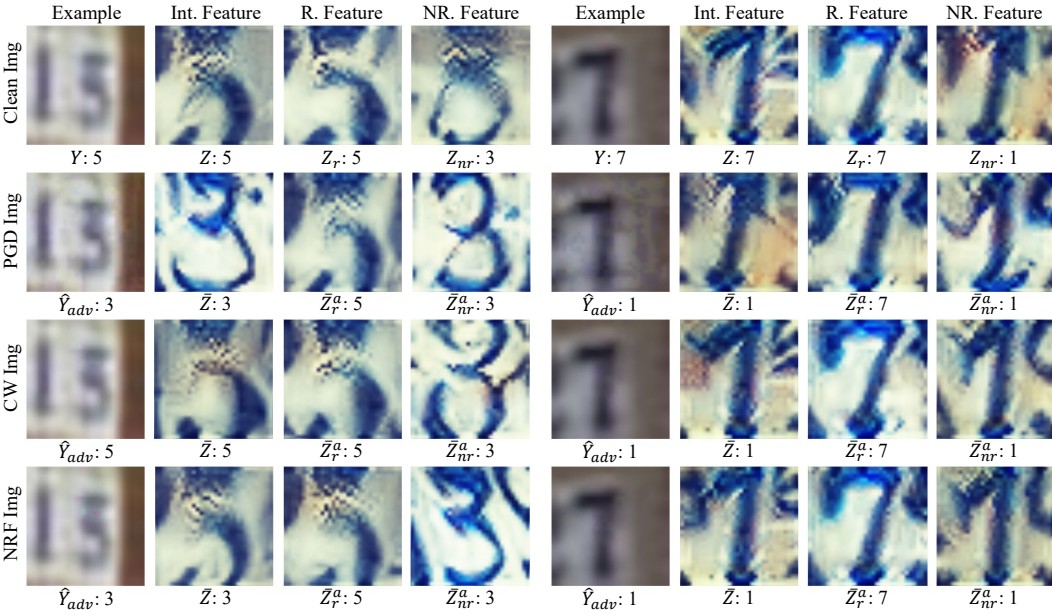

Figure 4: Feature visualization of SVHN. The class labels under each image indicate the target prediction results of the corresponding features.

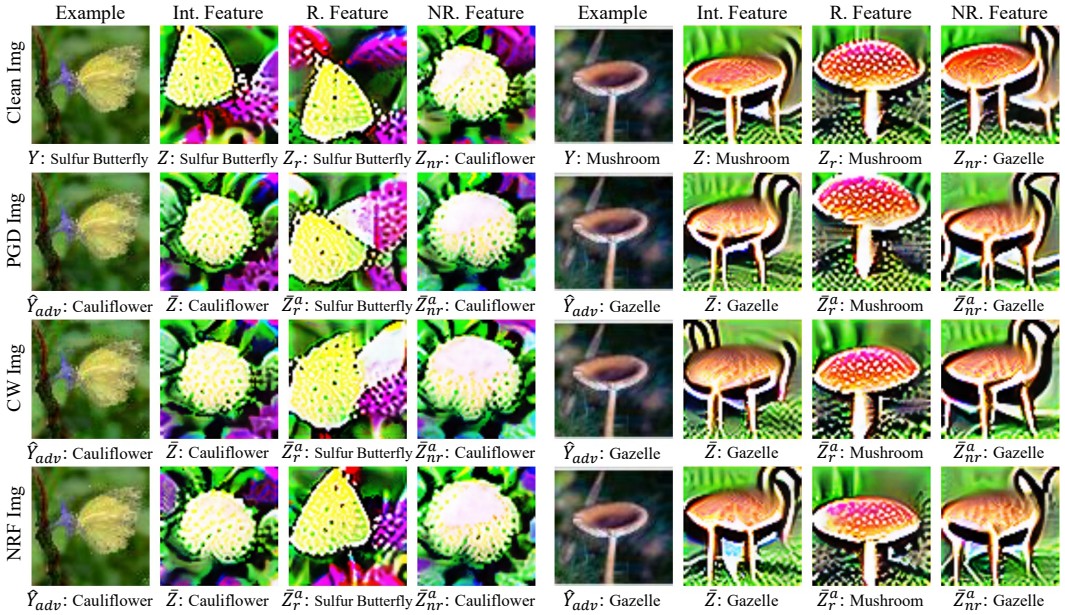

Figure 5: Feature visualization of Tiny-ImageNet. The class labels under each image indicate the target prediction results of the corresponding features.

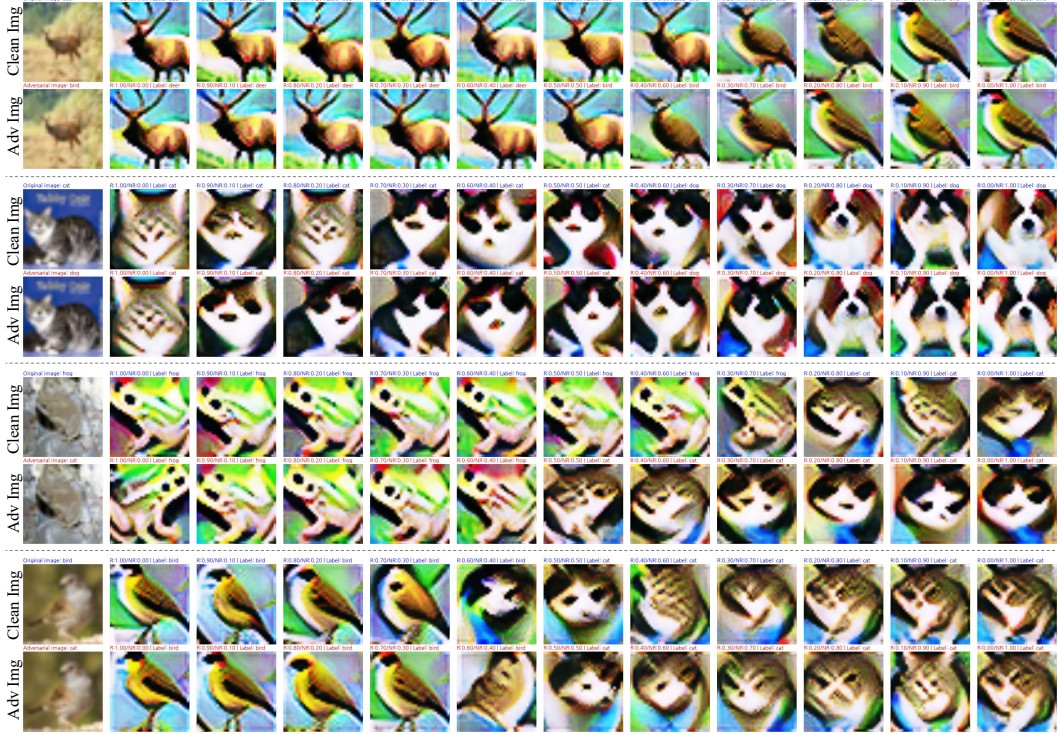

Figure 6: Interpolation results of feature visualization in CIFAR-10. We linearly interpolate the robust and non-robust features ranging of $\lambda = [0 : 0.1 : 1]$, and visualize the combination of feature representation $Z_{int}$. The ratio of the corresponding distilled features and predicted labels (*i.e.*, $f_{l+}(Z_{int})$) are described above the visualization examples.