# OpenReview forum: "Distilling Robust and Non-Robust Features in Adversarial Examples by Information Bottleneck"
_NeurIPS.cc/2021/Conference — NeurIPS 2021 Poster_

### Official Review · Reviewer_jAXE · 2021-07-12

**Rating:** 6
**Confidence:** 2

**Summary:**

This paper focuses on robust and non-robust features in adversarial examples. This paper proposes a method of distilling robust and non-robust features via information bottleneck. Through analysis, this paper has shown the high correlation between the distilled features and the adversarial prediction. Based on that, the authors design a new attack that intensifies the gradient of non-robust features.

**Limitations And Societal Impact:**

It is encouraged to discuss more how to utilize the distilled robust features.

**Main Review:**

Pros:

1. This paper proposes a new way to distill robust and non-robust features via information bottleneck.
2. The detailed method are well illustrated and sounds feasible.
3. Extensive analysis on the distilled features is provided and the figures or tables are well presented.
4. The current experimental results have shown the strong performance of the proposed attack.

Cons:

1. The motivation for using the information bottleneck seems to be unclear. Why we need the information bottleneck? Why information bottleneck can help to distill robust and non-robust features precisely? It is better to clearly discuss more in the former part of this paper.
2. It is better to discuss more the difference between this paper and [1] since the latter also provides a way to deal with robust and non-robust features. What is the relationship between the definitions in the two papers of robust/non-robust features? What is the unique significance of using information bottleneck to distill robust/non-robust features? Since the latter paper has already demonstrated the high correlation between the prediction and robust/non-robust features and proposed an easy but effective way to figure out these features.

Minor Concerns/Questions:

1. Can the authors verify the effectiveness of the proposed attack method on a larger dataset? Can the authors discuss the impact of dataset size on the performance of the attack method?

[1]. Ilyas, Andrew, et al. "Adversarial Examples Are Not Bugs, They Are Features." Advances in Neural Information Processing Systems 32 (2019): 125-136.

**Time Spent Reviewing:**

6

---

> ### Author Response · Authors · 2021-08-10
> **Responses for Reviewer jAXE**
>
> We appreciate the reviewer for the constructive comments and feedbacks. We took all the comments thoughtfully and regarded them as great opportunities to enhance the overall quality of our paper. We have addressed the reviewers questions below.
>
> ---
>
> **Q1. The motivation for using ... former part of this paper.**
>
> **A1.** To measure the prediction sensitivity along the noise variation directly manipulating the features, it is natural to adopt cross-entropy, a canonical form of evaluating the model prediction. In addition, when utilizing the variation of the features in an intermediate layer, it should be considered not to extremely deviate possible range of feature space. In other words. we should alleviate feature heterogeneity between original features $Z$ and the informative features $Z_{I}$ with the variation added to $Z$. Therefore, we need to compress the feature space to well represent highly predictive information even with a small variation. It is usually KL divergence that plays a role of the mentioned purpose. This is why we employ Information bottleneck which concurrently considers cross entropy and KL divergence in perspective of regulating and filtering neural network’s information flow through a notion of bottleneck structure. By using it, we can decompose the feature representation in a more detailed way.
>
> To clearly deliver the motivation of Information Bottleneck for distilling the robust and non-robust features, we will arrange this significant motivation in the former part of this paper.
>
> ---
>
> **Q2. It is better to discuss ... to figure out these features.**
>
> **A2.**  The definition of the robust and non-robust features are akin with the previous work [1] in that the robust features are literally robust and the non-robust features are vulnerable for the adversarial perturbation. But, Ilyas et al. [1] remark that it is difficult to directly manipulate the features of very complex, high-dimensional datasets. Alternatively, they construct the robust and non-robust datasets leveraging a robust model to modify dataset to contain the robust or non-robust features.
>
> Unlike their work, this paper proposes a novel way of directly decomposing the intermediate feature representations into the robust and non-robust features employing Information Bottleneck mechanism. Through a notion of bottleneck structure, we can explicitly address information flow of the intermediate feature space and estimate prediction sensitivity of each feature units along the injected noise variation to the feature. We further leverage the explicitly decomposed non-robust features' gradients and suggest utilizing them in the adversarial attack mechanism (NRF attack).
>
> Moreover, while the previous work [1] has argued that the non-robust features have incomprehensible property to human recognition, we further reveal that the non-robust features include semantic information by themselves, as well as they are highly correlated with the adversarially attacked labels. Based on the reviewer’s suggestion, we will revise Line 301-304 to provide additive explanations of main differences with related works.
>
> ---
>
> **Q3. Can the authors verify the effectiveness ... performance of the attack method?**
>
> **A3.** We agree that further validation on the larger dataset is important to clearly show the effectiveness of our proposed method. As noted for answer to **[Q3](https://openreview.net/forum?id=90M-91IZ0JC&noteId=Rnt48tnC26E)** of Reviewer CPc8, we show the below table to verify the effectiveness of the proposed attack method on a larger dataset, Tiny-ImageNet ($64\times 64$ pixels) which is a small subset of ImageNet dataset, separated in 200 different classes. Based on the experimental results of below table, non-robust feature (NRF) attack has significant performance on Tiny-ImageNet compared with other standard (FGSM, BIM, PGD, CW) and advanced attacks (AA, FAB).
>
> | Model | Method | Clean | FGSM | BIM | PGD | CW | AA | FAB | NRF |
> |------|--------|:-----:|:-----:|:-----:|:-----:|:-----:|:-----:|:-----:|:----:|
> | VGG | ADV | 34.0 | 17.5 | 16.5 | 16.1 | 12.0 | 15.4 | 12.2 | 6.7 |
> | VGG | TRADES | 38.7 | 20.1 | 19.1 | 18.7 | 13.9 | 17.8 | 13.3 | 7.8 |
> | VGG | MART | 38.4 | 20.6 | 19.5 | 19.1 | 14.1 | 18.3 | 14.7 | 9.2 |
> | WRN | ADV | 43.1 | 20.4 | 19.3 | 18.8 | 13.5 | 18.1 | 14.2 | 5.3 |
> | WRN | TRADES | 47.2 | 26.7 | 25.6 | 25.2 | 17.4 | 24.4 | 17.7 | 9.6 |
> | WRN | MART | 48.5 | 27.4 | 26.1 | 25.7 | 17.5 | 25.0 | 17.8 | 9.9 |
>
> In addition, we would like to thank for a suggestion to discuss the impact of dataset size on the performance of the attack method.
> The performance of the proposed attack is directly connected with well disentangling non-robust features in intermediate features. Normally, it is not easier to discern robust and non-robust features for large dataset in adversarial settings. This is because the larger dataset size, the harder it is for the neural network to learn the large dataset to achieve desired standard accuracy more than small datasets (CIFAR-10, SVHN). It also results in preventing neural networks from learning robust representation [2].
>
> However, in this paper, regardless of whether the dataset is large or small, we introduce the noise variation that can measure model’s sensitivity for the variation of model prediction. The reason why the noise variation does not depend on the dataset size is that our Information Bottleneck tries to find the noise variation which simply minimizes the cross entropy and KL divergence irrespective of dataset size in adversarial setting. Therefore, it can effectively help to distill the non-robust features that keep the proposed attack strong from any dataset. To better understand our contribution, we will arrange the above experimental results and discussion about dataset size, and attach them to our manuscript.
>
> ---
>
> References
>
> - [1] A. Ilyas, S. Santurkar, D. Tsipras, L. Engstrom, B. Tran, and A. Madry, “Adversarial examples are not bugs, they are features,” in Neural Information Processing Systems (NeurIPS), vol. 32, 2019.
> - [2] L. Engstrom, A. Ilyas, S. Santurkar, D. Tsipras, B. Tran, and A. Madry, “Adversarial robustness as a prior for learned representations,” arXiv preprint arXiv:1906.00945, 2019.

---

### Official Review · Reviewer_LQdF · 2021-07-16

**Rating:** 6
**Confidence:** 4

**Summary:**

This paper categorizes the intermediate features of a DNN into robust ones and non-robust ones. Through visualization, the non-robust ones display different semantic information from the true label, and thus potentially leads to the existence of AE.  The paper proposes a new AE generation algorithm targeting the non-robust feature, and beats the common attack baselines.

**Ethical Concerns:**

No concern.

**Limitations And Societal Impact:**

They are adequately addressed.

**Main Review:**

I like the overall idea of the paper. It delves deep into the potential cause of AE and shows an interesting perspective.

However, this paper suffers from the poor writing. There's no apparent grammar error, but the paper is almost un-readable because of wordy expressions, numerable inline definitions and lacking of concise algorithmic description. The algorithmic steps are buried in definition of terms and lengthy details. Here are some of my suggestions.

-- Present the definition of IB, the definition of robust and non-robust feature and the algorithm of distinguishing them in separate definition boxes or separate paragraphs. Never use inline definition for key terms.

-- Instead of immediately adding comments to the parameters in an equation, wait for the main content be defined and put analysis together. There's an analysis section, so the analysis should all go there... For each aspect of the analysis, add a paragraph header if subsection header takes too much space.

-- Make sure all terms in the key equations are defined. For example in Equation 3), the term f_{l+} is not defined.

-- Use concise expression. For example, starting from Line 97 "Here, ..." to the end of that paragraph, the entire part can be replaced by "Since finding the exact value of the second term is computationally prohibitive, we use the heuristics in [18, 19]." Save the space for **your** contribution.

-- Make the caption count. For example for the caption of Figure 3, explicitly say the non-robust feature displays semantics similar to the misclassified label of AE.

Overall, I like the contribution of this paper. However, in order for the community to better understand the paper and to more conveniently refer to the formulation in this work, I want to see a careful revision in writing. Therefore, I'm giving a score of 5 for now.

============ Post Rebuttal Response =======================

Thanks for the response. It's unfortunate that Neurips doesn't allow uploading an updated draft. I will raise my score to 6 and trust the authors to make another pass over the writing thoroughly.

**Time Spent Reviewing:**

3

---

> ### Author Response · Authors · 2021-08-10
> **Responses for Reviewer LQdF**
>
> We appreciate to the reviewer for acknowledging our contribution and thorough suggestions to improve the manuscript. We would like to update our improved revision manuscript, but unfortunately it is not allowed in this rebuttal period. Therefore, we will answer how we will reflect the reviewer’s writing suggestions in the next potential revision and discussion period.
>
> ---
>
> **Q1. Present the definition ... for key terms.**
>
> **A1.** We agree with that the many inline definitions potentially disturb to clearly understand our paper. We will make sure to separate paragraph for the key term based on the reviewer’s suggestion as follows. Moreover, we will move Algorithm 1 in Appendix A to the main body for better understanding of overall procedure of our method.
>
> Below notions are current inline definitions which we will revise to the separate paragraph or definition boxes.
> 1. In section 2
> + Line 100: Information bottleneck $\rightarrow$ $\min L_{CE}+\beta L_I$
> + Line 107: Informative features $\rightarrow$ $Z_{I}=f_l(X)+\sigma\cdot\epsilon$, where $Z_I\sim q_{\sigma}(Z)$
> + Line 122 - 125: Noise variation/Feature variation $\rightarrow$  $\sigma=[\sigma_1,\sigma_2,\cdots,\sigma_C]$ / $\sigma_Z=[\sigma_{Z_1},\sigma_{Z_2},\cdots,\sigma_{Z_C}]$
> + Line 151-152: Robust/Non-robust features  $\rightarrow$ $Z_{r}=i_{r} \cdot Z$ / $Z_{nr}=i_{nr} \cdot Z$
>
> 2. In section 4
> + Line 278: Objective of Non robust feature (NRF) attack $\rightarrow$ $\min\limits_{\delta} L_{base}-\|G_{nr}\|_{2}$
>
> ---
>
> **Q2. Instead of immediately adding ... header takes too much space**
>
> **A2.** As the reviewer commented, we will clarify the main contents first rather than immediately explaining parameters of equations in section 2.
> Furthermore, we will integrate the paragraph on the analysis of beta changes (Line 126 - 131) into the Line 239 in the section 3.4. To save space in analysis section, we will replace subsection header with paragraph header as the reviewer mentioned.
>
> ---
>
> **Q3. Make sure all terms ... term $f_{l+}$ is not defined.**
>
> **A3.** The term $f_{l+}$ is predefined in Line 78 to 80 of Problem Setup and Notations part, but we agree with that the all terms in key equations should clearly stand out. We embrace the reviewer’s suggestion and make sure to clearly define all important terms in equations.
>
> ---
>
> **Q4. Use concise expression ... the space for your contribution.**
>
> **A4.** We thank to reviewer’s suggestion. We will replace Line 97 to the end of the paragraph with “Since finding the exact value of the second term is computationally prohibitive, we use the heuristics in [18, 19].” and make sure to save space to describe our contributions in detail.
>
> ---
>
> **Q5. Make the caption count ... to the misclassified label of AE.**
>
> **A5.** We will make sure to revise the captions more clearly. Especially, for the caption of Fig.3, we will modify the explanations as: “Feature visualization [25] for the intermediate feature (Int.), robust feature (R.), and non-robust feature (NR.). The class labels under each image indicate the predicted results of the corresponding features propagated by $f_{l+}(\cdot)$. Note that the visualization of the non-robust features displays semantic similarity of the misclassified classes of the adversarial examples. Please see more visualization results in Appendix E.”
>
> ---
>
> We sincerely hope that this revision effort meets the reviewer’s expectation of providing better manuscript quality. We are happy to provide further clarification of manuscript in the next discussion period and we are willing to open the code to the public for conveniently referring our proposed method.

---

### Official Review · Reviewer_CPc8 · 2021-07-16

**Rating:** 6
**Confidence:** 3

**Summary:**

The paper proposes to explicitly separate intermediate representation into robust and non-robust categories by information bottleneck, and demonstrate that the non-robust feature is related to adversarial prediction. An attack mechanism based on enforcing non-robust features gradient is proposed. The experimental results show the effectiveness of the proposed approach.


**Limitations And Societal Impact:**

The authors have adequately addressed the limitations and potential negative societal impact of their work.

**Main Review:**

Information bottleneck(IB) has been widely used for feature disentanglement. It’s an interesting idea to connect IB to separate robust and non-robust features under adversarial settings.  Overall, the paper is well-organized. Hopefully, the authors can help address the following questions.


**1.** In line 119, how did you calculate the inherent feature variation $\sigma_z$? Is it obtained by computing the variance of the same unit along the batch size dimension within a batch? I think it needs to be clarified since its maximal value directly determines the robust and non-robust index.


**2.** Amplifying the brittleness of non-robust features is proposed as an attack. It is a typical white-box and untargeted attack, of which the applicable scenario is limited these days. Usually, we evaluate the effectiveness of an attack by attacking a baseline defense approach. There are many defense works against adversarial attacks[1]. To make the paper stronger, I think the authors should try at least one defense strategy to verify the proposed attack.


[1] [Towards Deep Learning Models Resistant to Adversarial Attacks.](https://arxiv.org/abs/1706.06083)



**3.**  The channel index acts like a mask to filter out non-robust features, which is highly dependent on the dataset used. On the other hand, most of the experiments are done on CIFAR-10 and SVHN, which both are small datasets and nicely distributed. I wonder about the transferability to larger datasets of the proposed method


**Time Spent Reviewing:**

3

---

> ### Author Response · Authors · 2021-08-10
> **Responses for Reviewer CPc8**
>
> We thank to the reviewer for acknowledging the idea of Information Bottleneck to decompose the robust and non-robust features under adversarial settings. Below, we would like to respond the remaining questions, and we will please to provide further clarification for the any additional questions in the next discussion period.
>
> ---
>
> **Q1. In line 119, ... the robust and non-robust index.**
>
> **A1.** For example, once we propagate an input $X$, an intermediate feature $Z=f_{l}(X)$ has $C \times H \times W$ shape. The shape of $\sigma_{Z}$ is $C \times 1 \times 1$, which is calculated with feature elements to each channel. Then, a maximum tolerance $T=\max \sigma_{Z}^2$ has one-dimension shape (scalar). To expand it, once we propagate input batch $X$ with size $B$, intermediate features $Z=f_{l}(X)$ have $B \times C \times H \times W$ shape, and the shape of $\sigma_{Z}$ is $B \times C \times 1 \times 1$. In each batch, it is calculated with feature elements to each channel. Then, maximum tolerances $T=\max \sigma_{Z}^2$ have $B$ dimension (batch size).
>
> To clearly deliver the procedures of distilling robust/non-robust features by our Information Bottleneck, we will definitely add the above explanation for the dimension of the noise variation $\sigma$, the feature variation $\sigma_{Z}$, and the maximum tolerance $T$ along batch size.
>
> ---
>
> **Q2. Amplifying the brittleness ... verify the proposed attack.**
>
> **A2.** Since our approach directly manipulates feature representation and decomposes the robust and non-robust features, it is necessary to access internal information of the neural network. Therefore, we have assumed white-box settings to amplify the brittleness of the non-robust features by utilizing the gradient of them, and we have verified the effectiveness of non-robust feature (NRF) attack on the white-box settings.
>
> Moreover, we agree with the reviewer’s comment on the comparison with other recent defense strategies. We kindly inform that as described in Table 3, we have already compared “ADV” (PGD adversarial training) as well as advanced defense methods (“TRADES” [1] and “MART” [2]) to clearly validate the performance of NRF attack.
>
> ---
>
> **Q3. The channel index acts ... of the proposed method.**
>
> **A3.** To demonstrate the effectiveness of the proposed method, in below table, we show the performance of NRF attack in a larger dataset, Tiny-ImageNet ($64\times 64$ pixels) which is a small subset of ImageNet dataset, separated in 200 different classes. Based on the experimental results of below table, non-robust feature (NRF) attack has significant performance improvements even on Tiny-ImageNet compared with other standard (FGSM, BIM, PGD, CW) and advanced attacks (AA, FAB). Furthermore, as illustrated in Table 1-2, Figure 3, the analyses of Tiny-ImageNet are totally aligned with those of the small datasets: CIFAR-10 and SVHN.
>
> | Model | Method | Clean | FGSM | BIM | PGD | CW | AA | FAB | NRF |
> |------|--------|:-----:|:-----:|:-----:|:-----:|:-----:|:-----:|:-----:|:----:|
> | VGG | ADV | 34.0 | 17.5 | 16.5 | 16.1 | 12.0 | 15.4 | 12.2 | 6.7 |
> | VGG | TRADES | 38.7 | 20.1 | 19.1 | 18.7 | 13.9 | 17.8 | 13.3 | 7.8 |
> | VGG | MART | 38.4 | 20.6 | 19.5 | 19.1 | 14.1 | 18.3 | 14.7 | 9.2 |
> | WRN | ADV | 43.1 | 20.4 | 19.3 | 18.8 | 13.5 | 18.1 | 14.2 | 5.3 |
> | WRN | TRADES | 47.2 | 26.7 | 25.6 | 25.2 | 17.4 | 24.4 | 17.7 | 9.6 |
> | WRN | MART | 48.5 | 27.4 | 26.1 | 25.7 | 17.5 | 25.0 | 17.8 | 9.9 |
>
> ---
>
> References
>
> - [1] H. Zhang, Y. Yu, J. Jiao, E. Xing, L. El Ghaoui, and M. Jordan, “Theoretically principled trade-off between robustness and accuracy,” in International Conference on Machine Learning (ICML), pp. 7472–7482, PMLR, 2019.
> - [2] Y. Wang, D. Zou, J. Yi, J. Bailey, X. Ma, and Q. Gu, “Improving adversarial robustness requires revisiting misclassified examples,” in International Conference on Learning Representations (ICLR), 2020.

---

### Official Review · Reviewer_Yvhe · 2021-07-16

**Rating:** 8
**Confidence:** 3

**Summary:**

This work utilizes an Information Bottleneck (IB) to distill robust and non-robust features in neural networks (NNs). Experiments are performed to show that the distilled features are correlated with adversarial prediction and are human-perceptible. Finally, an attack mechanism intensifying the gradient of non-robust features is proposed.

**Limitations And Societal Impact:**

Please refer to the main review.

**Main Review:**

Utilizing IB to capture relevant information is not a novel technique, but distilling robust and non-robust features in the intermediate layers of networks with it sounds interesting. The authors define the robustness and non-robustness of units by their correlation to the model prediction and separate informative features into robust and non-robust units by their tolerance of feature variation, which is a novel perspective to study the robustness of NNs. Experiments along with visualizations can well support their claims. There is also question: What is the connection between the robustness and informativeness? Does Figure 2 implies that non-robust features are less informative than robust features?

**Time Spent Reviewing:**

5

---

> ### Author Response · Authors · 2021-08-10
> **Responses for Reviewer Yvhe**
>
> We appreciate that the reviewer acknowledged the novelty of our method, as well as new perspective of studying the robustness in DNNs. We expect to deploy our definition of robust and non-robust features to spur the future research of adversarial robustness. Below, we would like to address the remaining questions of the reviewer.
>
> ---
>
> **Q1. What is the connection between the robustness and informativeness? Does Fig. 2 implies that non-robust features are less informative than robust features?**
>
> **A1.** Based on our experimental results, we can infer the relation between the robustness and informativeness of the distilled features. For the experiments on the clean examples in Table 1, we have observed that the non-robust features seemingly derive moderate prediction results (e.g., 34.82%, 57.09% in CIFAR-10 and SVHN datasets), of which t-SNE plot somewhat shows clustered results in Fig 2(a). Whereas, the robust features clearly show consistent predictive accuracy and well-clustered t-SNE results even in the adversarial perturbation. Therefore, we can say that the robust features may have relatively more informativeness than the non-robust feature for the “target class”, as the reviewer commented.
>
> One thing we would like to notice is that we observe that the non-robust features of the adversarial examples are highly correlated with the adversarially attacked class, as analyzed in section 3.3. Therefore, when the adversarial perturbation exists, we can infer that the non-robust features lose informativeness of the target class and enhance manipulated information of the attacked class, while the robust features retain invariance of informativeness for the target class.

---

### Official Review · Reviewer_xNdo · 2021-07-17

**Rating:** 6
**Confidence:** 3

**Summary:**

This study aims at distilling feature representations (mainly the last convolution layer) into robust and non-robust features using information bottleneck. For every layer, the feature maps are partitioned into robust and non-robust for each individual example. After finding this, the paper first illustrates that selectively propagating robust features of adversarial examples built for a robust network will not degrade the accuracy but propagating non-robust features does which to some extent illustrates that the information bottleneck method has correctly identified the robust and non robust features. They also illustrate that the adversarial class label has high correlation to the Non-robust features.


**Main Review:**

The paper has some interesting insights but they all correspond to what is expected and it doesn’t provide much new understanding of robustness. Also, the paper's clarity can be significantly improved, for example, there are parts which are referenced before and then introduced after (for example, Z^a_{nr} is referenced in section 3.2 and figure 3 but is defined for the first time in section 3.3). After reading the paper with care, I am still not sure if I understood the experiments completely. The paper extends the idea of input data having robust and non-robust features to deeper “learned” representations of a Neural Net. While in the original adversarial examples are features not bugs paper, the robust and non-robust features co-existed in each channel, in this paper the authors classify each channel as robust or non-robust. I do wonder whether more straightforward methods exist for this partitioning purpose — possibly, simply using the ell_2-norm of the gradient of the loss w.r.t. the feature map —  which can result in similar qualitative results?

One interesting factor seems to be from the fact that the non-robust deep features are also perceptually aligned. However, I am not sure if this is true in general, i.e., when we do not use regularizations used for visualizations and we partition deep features for naturally trained models. Do the authors have an intuition for that case? I do wonder if robust and non robust features of a naturally trained net are also perceptually aligned? The finding that the non-robust feature's visualizations for a natural example on an adversarially trained model visually look like the adversarial class is not shocking and is expected due to the perceptual alignment of robust networks where if you make an adversarial example with a large epsilon for them, the adversarial example looks like an image from the adversarial class.

I do have the following other questions which could help me understand if I understood the paper good or not:

* Is there a reason for using a robust net in Table1? Does the network have to be adversarially trained to have clearly separable R and NR features?
* The NRF attack does not seem to be ell-infitiny bounded, is this correct?  If yes, comparisons of that with other attacks in table 3 does not seem to be fair.

**Time Spent Reviewing:**

5

---

> ### Author Response · Authors · 2021-08-10
> **Responses for Reviewer xNdo**
>
> We would like to thank the reviewer for the construct feedback and thoughtful suggestions that we should take into account. Below, we have inline answers for the reviewer’s questions.
>
> ---
>
> **Q1. Also the paper’s clarity can be … defined for the first time in section 3.3).**
>
> **A1.** We thank the reviewer for the comments regarding clarity of our manuscript. We will ensure to move the definition of $\bar{Z}^a_r$ and $\bar{Z}^a_{nr}$ to the section 3.2 from section 3.3 to reflect the reviewer’s comment and make it clear for better understanding.
>
> ---
>
> **Q2. After reading the paper with care … in similar qualitative results?**
>
> **A2.** Ilyas et al. [1] have defined that “a feature to be a function mapping from the input space to the real numbers”. Also, they have regarded the features as abstractly sematic information and categorized the features as robust or non-robust. In general CNN architecture, various works have demonstrated that the semantic information is included in the feature units (each channel in feature representation) as mentioned in Line 88. Therefore, we would like to kindly point out that it is difficult to assure that the robust and non-robust features co-exist in each channel, and it is reasonable ground to set the feature units as each channel of the feature map.
>
> As the reviewer commented, we can estimate the gradient of the loss w.r.t the feature map, but the solely acquired norm of gradient loss does not explicitly perturb the feature representation. Thus, it is difficult to track the prediction sensitivity using the norm of gradient loss when adversarial perturbation exists. The main goal of using Information Bottleneck is measuring prediction sensitivity of the certain feature units against the injected noise variation to each feature unit. By optimizing the variation term in the bottleneck structure, we compare the noise variation in the informative features and decompose them to the robust or non-robust features based on the defined criterion (maximum tolerance).
> Also, we would like to note that a certain criterion needs to be predefined to identify whether each feature unit is robust or non-robust for the adversarial perturbation. We look forward to a variety of future works to define the criterion and novel approaches for decomposing intermediate feature representation directly.
>
> ---
>
> **Q3. One interesting factor seems to … are also perceptually aligned?**
>
> **A3.** We can refer [2] which argued that pairs of images that appear completely different can correspond to similar feature representation in standard setting. Thus, feature visualization fails for the standard networks (In adversarial setting, the robust feature representations are invertible). As described in [2], while the feature representations of robust networks include human-recognizable information, the one from the standard networks cannot show visual improvements even with advanced regularization techniques. Therefore, we visualize the robust and non-robust features in the adversarially trained networks. Moreover, it is even inaccessible to decompose the feature representation in the standard networks using prediction sensitivity. We will explain more details why it is difficult to decompose feature representation of standard networks in the answer of [Q5](https://openreview.net/forum?id=90M-91IZ0JC&noteId=aB4bjrOzCF4).
>
> In this paper, we utilize Information Bottleneck structure and clearly show that the decomposed robust and non-robust features estimated by noise variation represent semantic information by themselves in adversarial setting. Note that our comparison of the robust and non-robust feature visualization results for the clean and adversarial examples, are conducted on the same conditions of visualization method [4], and both show perceptible semantic information as illustrated in Fig. 2.
>
> ---
>
> **Q4. The finding that the non-robust … from the adversarial class.**
>
> **A4.** We agree with the reviewer’s comment that the generated adversarial example with large epsilon may look like an image of adversarially attacked class in input space as discovered in [3]. However, we would like to kindly note that the adversarial perturbation is currently set as imperceptible noise to human eyes, thus the adversarial examples look similar to the given input images. In this general set-up as default, the previous work [1] has argued that the non-robust features are not human-aligned and include unrecognizable information. Note that we have revealed that each robust and non-robust feature decomposed by our Information Bottleneck mechanism includes semantic information in the intermediate feature space (not in input space) with imperceptible noise. Especially, we would like to point out that the visual results of the non-robust features are highly correlated with the adversarial class. We believe this observation contributes a way of interpreting the cause of adversarial examples and present a novel perspective of studying robustness in DNNs. We hope that the additional visualization results and visual interpolation in the feature space (Appendix E) provide better understanding of how the non-robust features subtly manipulate the model prediction to the adversarial prediction.
>
> ---
>
> **Q5. Is there a reason for using a robust net in Table 1? Does the network have to be adversarially trained to have clearly separable R and NR features?**
>
> **A5.** In standard settings, feature representations are intensely vulnerable for the adversarial perturbation [1]. Thereby, injecting noise variation to the standard feature representations can be easily manipulated and it can flip the model prediction. Therefore, the robustness of each feature unit cannot be estimated in standard networks.
>
> It also corresponds with the observation of the previous work [1] that have demonstrated the distinction between robust and non-robust features arises in adversarial setting. We want to kindly inform the reviewer that we put an endnote in page 2 to explain why we experiment and analyze the robust and non-robust features in the adversarial setting. Moreover, we have attached careful analysis of our proposed method for the canonical standard networks in Appendix F. As in the analysis, we have clearly showed that the non-robust features are not solely brittle, but highly predictive concurrently in standard setting. Thus, it is reasonable ground that the standard setting is not suitable for analyzing robust and non-robust features.
>
> ---
>
> **Q6. The NRF attack does not seem to be ell-infinity bounded, is this correct? If yes, comparisons of that with other attacks in Table 3 does not seem to be fair.**
>
> **A6.** We also agree with reviewer’s concern on the fair comparison, and we would like to clearly explain that we validate our proposed attack in the fair condition. In Table 3, we compare FGSM, BIM, PGD, AA, FAB (Linf) with CW, NRF (L2). In fact, NRF is a type of L2 optimization-based attack which belongs to CW attack [5]. Here, main motivation of L2 optimization-based attack such as CW is to make stronger attack than Linf attack for defensive distillation  [5, 6, 7] with much less change of the original image by using “change of variables” on L2 metric. From this perspective, to compare with stronger attack performance, many research [8, 9, 10] have employed CW-L2 to verify the robustness in the model. Therefore, in this paper, we also use a more powerful attack yet with less changes to the original image, NRF-L2.
> Furthermore, as described in validation part (Appendix A: Implementation Detail), we have mentioned for NRF attack that “we equally use hyperparameters of CW with L2 distance metric of 0.01 to fairly validate its effectiveness for intensifying brittleness of non-robust features.”
>
> ---
>
> References
> - [1] A. Ilyas, S. Santurkar, D. Tsipras, L. Engstrom, B. Tran, and A. Madry, “Adversarial examples are not bugs, they are features,” in Neural Information Processing Systems (NeurIPS), vol. 32, 2019.
> - [2] L. Engstrom, A. Ilyas, S. Santurkar, D. Tsipras, B. Tran, and A. Madry, “Adversarial robustness as a prior for learned representations,” arXiv preprint arXiv:1906.00945, 2019.
> - [3] D. Tsipras, S. Santurkar, L. Engstrom, A. Turner, and A. Madry, “Robustness may be at odds with accuracy,”
> 358 in International Conference on Learning Representations (ICLR), 2019.
> - [4] C. Olah, A. Mordvintsev, and L. Schubert, “Feature visualization,” Distill, vol. 2, no. 11, p. e7, 2017.
> - [5] N. Carlini and D. Wagner, “Towards evaluating the robustness of neural networks,” in 2017 IEEE Symposium on Security and Privacy (SP), pp. 39–57, IEEE Computer Society, 2017.
> - [6] Olga Taran, Shideh Rezaeifar, Taras Holotyak, Slava Voloshynovskiy, "Defending Against Adversarial Attacks by Randomized Diversification", Proceedings of the IEEE/CVF Conference on Computer Vision and Pattern Recognition (CVPR), 2019, pp. 11226-11233
> - [7] Eniser, H. F., Christakis, M., & Wüstholz, V. Raid: Randomized adversarial-input detection for neural networks. arXiv preprint arXiv:2002.02776.
> - [8] Gilad Cohen, Guillermo Sapiro, Raja Giryes, “Detecting Adversarial Samples Using Influence Functions and Nearest Neighbors” Proceedings of the IEEE/CVF Conference on Computer Vision and Pattern Recognition (CVPR), 2020, pp. 14453-14462
> - [9] Wieland Brendel, Jonas Rauber, Matthias Kümmerer, Ivan Ustyuzhaninov, Matthias Bethge, “Accurate, reliable and fast robustness evaluation”, in Neural Information Processing Systems (NeurIPS), vol. 32, 2019.
> - [10] Yi-Hsuan Wu, Chia-Hung Yuan, Shan-Hung Wu, “Adversarial Robustness via Runtime Masking and Cleansing”, Proceedings of the 37th International Conference on Machine Learning (ICML), PMLR 119:10399-10409, 2020.

---

> > ### Comment · Reviewer_xNdo · 2021-09-03
> > **Thank you for providing clarification -- the presentation can be improved**
> >
> > Thanks for the rebuttal. After reading it carefully, I will increase my score by 1 point. I believe the work is fine, however, the results are not unexpected and the presentation is not ideal these are the main reasons which prevent me from giving it an accept score in its current form. In addition, I don't see how clearly one can use these findings to their benefit due to very few fair comparisons and examples. The main example provided in the paper is to use NRF for crafting strong adversarial examples. However, that itself is not very clear from Table 3 as there are comparisons to methods which are not comparable (L-inf vs L-2). The NRF attack is not an $\ell_{\infty}$ bounded attack and quite frankly it is not explicitly mentioned anywhere in the main body of the paper that the CW attack is an CW-L2 attack -- I suggest moving that information to the main body; In fact I suggest that Table 3, should only contain L-2 attacks (L-2 PGD, L-2 AA, etc.).

---

> > > ### Author Response · Authors · 2021-09-08
> > > **Responses for Reviewer xNdo**
> > >
> > > We would like to thank the reviewer for suggestions to improve the presentation in this paper. Below, we have inline answers for the reviewer’s suggestions.
> > >
> > > ---
> > >
> > > **Q1. I suggest moving that information to the main body.**
> > >
> > > **A1.** We agree with that it is better to move information of L2 bounded attack for CW and NRF attack to main body of this paper to clarify the experiment. Therefore, we will move validation part of implementation details in appendix A to main body of this paper.
> > >
> > > ---
> > >
> > > **Q2. In fact I suggest that Table 3, should only contain L-2 attacks (L-2 PGD, L-2 AA, etc.).**
> > >
> > > **A2.** To respond to the reviewer’s suggestion, we compared L2 attack performance of NRF with that of PGD, CW, AutoAttack, and FAB with $\epsilon=0.5$ attack magnitude. To fairly conduct the experiment, we limit actual L2 distance of adversarial perturbation generated by all of the attacks to 0.01. Below tables represent the performances of L2 bounded attacks on CIFAR-10, SVHN, Tiny-ImageNet.
> > >
> > > ---
> > >
> > > * **CIFAR-10**
> > >
> > > | Model | Method | Clean | PGD-L2 | CW-L2 | AA-L2 | FAB-L2 | NRF-L2 |
> > > |:-----:|--------|:-----:|:------:|:-----:|:-----:|:------:|:------:|
> > > |  VGG  | ADV    | 79.7  |  57.9  | 40.3  | 57.4  |  55.1  |  27.4  |
> > > |  VGG  | TRADES  | 78.2  |  62.0  | 43.0  | 61.1  |  58.0  |  31.2  |
> > > |  VGG  | MART   | 73.5  |  60.5  | 42.2  | 60.2  |  56.4  |  31.4  |
> > > |  WRN  | ADV    | 82.6  |  60.9  | 45.5  | 60.7  |  57.2  |  17.1  |
> > > |  WRN  | TRADES  | 83.0  |  61.5  | 46.7  | 61.3  |  57.7  |  26.8  |
> > > |  WRN  | MART   | 83.4  |  60.7  | 46.5  | 60.8  |  57.8  |  19.6  |
> > >
> > > ---
> > >
> > > * **SVHN**
> > >
> > > | Model | Method | Clean | PGD-L2 | CW-L2 | AA-L2 | FAB-L2 | NRF-L2 |
> > > |:-----:|--------|:-----:|:------:|:-----:|:-----:|:------:|:------:|
> > > |  VGG  | ADV    | 90.4  |  34.6  | 33.3  | 33.0  |  25.5  |  12.6  |
> > > |  VGG  | TRADES | 90.4  |  47.5  | 44.8  | 46.5  |  44.5  |  14.3  |
> > > |  VGG  | MART   | 90.5  |  47.9  | 46.4  | 46.7  |  42.2  |  16.1  |
> > > |  WRN  | ADV    | 93.5  |  46.1  | 40.7  | 45.8  |  37.3  |  13.4  |
> > > |  WRN  | TRADES | 93.9  |  47.6  | 42.0  | 47.3  |  45.6  |  10.1  |
> > > |  WRN  | MART   | 94.1  |  48.2  | 42.3  | 47.7  |  43.4  |  8.1   |
> > >
> > > ---
> > >
> > > * **Tiny-ImageNet**
> > >
> > > | Model | Method | Clean | PGD-L2 | CW-L2 | AA-L2 | FAB-L2 | NRF-L2 |
> > > |:-----:|--------|:-----:|:------:|:-----:|:-----:|:------:|:------:|
> > > |  VGG  | ADV    | 34.0  |  28.5  | 12.0  | 27.4  |  24.7  |  6.7   |
> > > |  VGG  | TRADES | 38.7  |  31.8  | 13.9  | 31.3  |  29.2  |  7.8   |
> > > |  VGG  | MART   | 38.4  |  32.2  | 14.1  | 32.6  |  29.1  |  9.2   |
> > > |  WRN  | ADV    | 43.1  |  33.1  | 13.5  | 33.2  |  30.5  |  5.3   |
> > > |  WRN  | TRADES | 47.2  |  39.8  | 17.4  | 39.1  |  36.9  |  9.6   |
> > > |  WRN  | MART   | 48.5  |  40.0  | 17.5  | 39.7  |  37.2  |  9.9   |

---

### Public Comment · ~Mingda_Zhang2 · 2021-12-20
**The variational distribution in this paper is not similar to the definition of variational information bottleneck.**

I think your results are very novel and nice. However, I have two questions about formulation when I read this paper.

1. The definition of $q_{\sigma}(Z)$ in your paper is related to $X$, which is not similar to the definition of variational information bottleneck. The traditional definition of variational distribution is not related to $X$, because they want to estimate the prior distribution of $Z$, $p(Z)$.  Is the definition in the paper reasonable?

2. The $\sigma_{z}$ is the internal channel variance of each sample. The traditional variance information bottleneck is learned from the neural network? Is there some related work which has computed the variance by every single image in the channels?

---

> ### Public Comment · ~Byung-Kwan_Lee1 · 2021-12-20
> **Thanks for the nice comments on the definition of the variational information bottleneck used in this paper.**
>
> Thanks for the nice comments on the definition of the variational information bottleneck used in this paper.
>
> ---
>
> **Q1. The definition of $q_{\sigma}(Z)$  in your paper is related to $X$, which is not similar to the definition of variational information bottleneck. The traditional definition of variational distribution is not related to $X$, because they want to estimate the prior distribution of $Z$, $p(Z)$. Is the definition in the paper reasonable?**
>
> **A1.** We agree that the traditional variational information bottleneck [1] using $q(Z)$ is not similar to the definition of our information bottleneck using $q_{\sigma}(Z)$. However, the traditional intention of variational information bottleneck [1] helps to embed intermediate features related to the target class yet concisely representing input. In addition, it has been well-known that it can allow deep neural networks to be learned to predict and generalize well for total datasets (e.g., variational autoencoders, variational dropout, and etc.), where it is assumed that $q(Z)=\mathcal{N}(0, I)$.
>
>
> On the other hand, the intention of our information bottleneck mechanism is not to learn the parameters of deep neural networks for prediction well to total datasets, but to analyze and manipulate intermediate features in adversarial settings given 'an' image, not for generalization to total datasets. Therefore, we provide $q_{\sigma}(Z)$ with prior information $f_{l}(X)$ as a form of the mean 'instead of zero' to sample highly predictive features for 'an' image $X$. For the mathematical perspective, what we can compute is not $p(Z)$ but $p(Z\mid X)$, what we should optimize is $D_{KL}( p(Z \mid X) \mid\mid q_{\sigma}(Z) )$, and then what we should learn is $\sigma$ in $q_{\sigma}(Z)$. Beyond this paper, furthermore, its procedure is also well represented in previous works [2]. Hence, we can say that our information bottleneck mechanism is reasonably defined.
>
> ---
>
> **Q2. The $\sigma_{Z}$ is the internal channel variance of each sample. The traditional variance information bottleneck is learned from the neural network? Is there some related work which has computed the variance by every single image in the channels?**
>
> **A2.** Equal to our work, the traditional Information Bottleneck [1] uses numerical mean and covariance of the intermediate features such that $p(Z \mid X)=\mathcal{N}(Z \mid f_{e}^{\mu}(X), f_{e}^{\Sigma}(x))$, where $f_{e}$ indicates a specific layer. (You can refer to equation 16 in [1]). There is a work [2] that handles intermediate features with their means and variances injected to measure attributions that return an upper bound on the amount of information.
>
> ---
>
> **References**
>
> [1] A. Alemi, I. Fischer, J. Dillon, and K. Murphy, “Deep variational information bottleneck,” in International Conference on Learning Representations (ICLR), 2017.
>
> [2] K. Schulz, L. Sixt, F. Tombari, and T. Landgraf, “Restricting the flow: Information bottlenecks for
> attribution,” in International Conference on Learning Representations (ICLR), 2019.

---

### Public Comment · ~Zhaoxi_Zhang1 · 2022-01-14
**It seems that Information Bottleneck do not take effect in distilling robust/non-robust feature.**

In this paper, robust/non-robust features are distilled by adding noise, this method will still work even if information bottleneck do not exist in loss function. In other words, what role does information bottleneck play in this process? Do you have further ablation study to extimate the effectiveness of information bottleneck in distilling?

---

> ### Public Comment · ~Junho_Kim4 · 2022-01-15
> **Answer for the important role of information bottleneck.**
>
> We have dealt with the important role of information bottleneck in the perspective of information flow through section 2.1. Also, the ablation study have conducted in Fig. 5. Please see section 3.4 for better understanding.

---

### Decision · Program_Chairs · 2021-09-27

**Decision:**

Accept (Poster)

**Comment:**

This paper utilizes the information bottleneck mechanism to disentangle robust and non-robust features in the representation space of a deep neural network. The paper shows a high correlation between the (wrong) prediction on an adversarial example and the identified non-robust features for the example. Finally, the paper proposes a new adversarial attack that aims to enlarge the gradient for the non-robust features to degrade the model performance.

Even though robust and non-robust features have been discussed in the context of adversarial examples before, this paper proposed an interesting and novel approach to disentangle these features in the representation space. The experiments in the paper do demonstrate a clear connection between the non-robust features and the performance of the model on adversarial examples. Furthermore, the identified adversarial attack is shown to be very effective in rendering the model useless. Thus, this paper will be a valuable addition to NeurIPS 2021.

There is some room for improvement in the writing of the paper. Please address the following issues in the revised version. Instead of formally introducing many notations, key definitions, and algorithmic details, they appear as part of various paragraphs. There is no discussion of the computational complexity issues, especially in the context of the proposed attack. Furthermore, how feasible is it to perform the proposed attack (NRF) as it requires first disentangling the robust vs. non-robust features and then accordingly perturbing those? Does NRF provide any guidance to train more robust models? Interestingly, it appears from Table 4, that model degradation is quite significant even when robust features are perturbed. In fact, perturbing the robust features is more effective than all baselines considered in Table 3 (obviously, except NRF). This point is glossed over in lines 284-288. Addressing these aforementioned issues will certainly improve the overall quality and impact of this paper.